# Endosomal phosphatidylserine is critical for the YAP signalling pathway in proliferating cells

Tatsuyuki Matsudaira[1], Kojiro Mukai[1], Taishin Noguchi[1], Junya Hasegawa[1], Tomohisa Hatta[2], Shun-ichiro Iemura[3], Tohru Natsume[2], Norio Miyamura[4], Hiroshi Nishina[4], Jun Nakayama[5], Kentaro Semba[5], Takuya Tomita[6], Shigeo Murata [6], Hiroyuki Arai[1,7,8] & Tomohiko Taguchi [1,7]

Yes-associated protein (YAP) is a recently discovered growth-promoting transcription coactivator that has been shown to regulate the malignancy of various cancers. How YAP is regulated is not fully understood. Here, we show that one of the factors regulating YAP is phosphatidylserine (PS) in recycling endosomes (REs). We use proximity biotinylation to find proteins proximal to PS. Among these proteins are YAP and multiple proteins related to YAP signalling. Knockdown of ATP8A1 (an RE PS-flippase) or evectin-2 (an RE-resident protein) and masking of PS in the cytoplasmic leaflet of membranes, all suppress nuclear localization of YAP and YAP-dependent transcription. ATP8A1 knockdown increases the phosphorylated (activated) form of Lats1 that phosphorylates and inactivates YAP, whereas evectin-2 knockdown reduces the ubiquitination and increased the level of Lats1. The proliferation of YAP-dependent metastatic cancer cells is suppressed by knockdown of ATP8A1 or evectin-2. These results suggest a link between a membrane phospholipid and cell proliferation.

[1] Department of Health Chemistry, Graduate School of Pharmaceutical Sciences, University of Tokyo, 7-3-1, Hongo, Bunkyo-ku, Tokyo 113-0033, Japan. [2] Molecular Profiling Research Center for Drug Discovery, National Institute of Advanced Industrial Science and Technology, 2-3-26, Aomi, Koto-ku, Tokyo 135-0064, Japan. [3] Medical Industry Translational Research Center, Fukushima Medical University, 1, Hikarigaoka, Fukushima 960-1295, Japan. [4] Department of Developmental and Regenerative Biology, Medical Research Institute, Tokyo Medical and Dental University, 1-5-45, Yushima, Bunkyo-ku, Tokyo 113-8519, Japan. [5] Department of Life Science and Medical Bioscience, School of Advanced Science and Engineering, Waseda University, 2-2, Wakamatsu-cho, Shinjuku-ku, Tokyo 162-8480, Japan. [6] Laboratory of Protein Metabolism, Graduate School of Pharmaceutical Sciences, The University of Tokyo, 7-3-1, Hongo, Bunkyo-ku, Tokyo 113-0033, Japan. [7] Pathological Cell Biology Laboratory, Graduate School of Pharmaceutical Sciences, The University of Tokyo, 7-3-1, Hongo, Bunkyo-ku, Tokyo 113-0033, Japan. [8] AMED-CREST, Japan Agency for Medical Research and Development, 1-7-1, Otemachi, Chiyoda-ku, Tokyo 100-0004, Japan. Tatsuyuki Matsudaira and Kojiro Mukai contributed equally to this work. Correspondence and requests for materials should be addressed to H.A. (email: harai@mol.f.u-tokyo.ac.jp) or to T.T. (email: tom_taguchi@mol.f.u-tokyo.ac.jp)

Membrane lipids not only serve as a physical barrier, but also interact with a wide variety of integral and peripheral membrane proteins to regulate their localization and activity[1]. Cellular membranes including the plasma membrane (PM) and the membranes of intracellular organelles have distinct lipid compositions[2, 3]. PS, a relatively minor constituent of biological membranes, is enriched in the inner leaflet of the PM[4, 5], and facilitates various signalling events through membrane translocation and activation of various kinases[6]. PS is also highly enriched in the cytoplasmic leaflet of REs[7, 8], and is important for endosomal membrane traffic[7, 9]. However, whether endosomal PS participates in intracellular signalling remains unclear.

Proximity-dependent biotin identification (BioID) is a recently developed method to identify protein–protein interactions[10]. The method is based on proximity-dependent cellular biotinylation by a promiscuous bacterial biotin ligase[11, 12] (*Escherichia coli* BirA R118G, hereafter called BirA*) fused to a bait protein. Biotinylated proteins can be selectively isolated by biotin capture methods and identified using mass spectrometry analysis[10, 13–15]. An advantage of BioID over conventional biochemical analyses is that it can identify transient or weak protein–protein interactions in vivo[10].

Here, we use BioID to identify proteins in close proximity to PS-enriched membranes. For the bait protein, we use a tandemly connected pleckstrin homology (2xPH) domain of evectin-2 that specifically binds PS and predominantly targets REs[7, 9]. As a result, we identify YAP, a critical growth-promoting transcription coactivator, as a PS-proximity protein. We also find that endosomal PS has a role in the YAP signalling pathway in proliferating cells.

## Results

**Identification of proteins in close proximity to PS.** COS-1 cells were stably expressed with a construct consisting of 2xPH, BirA* (a promiscuous biotin ligase that biotinylates proteins within a distance of 30 nm in the presence of biotin), and GFP (Fig. 1a). As

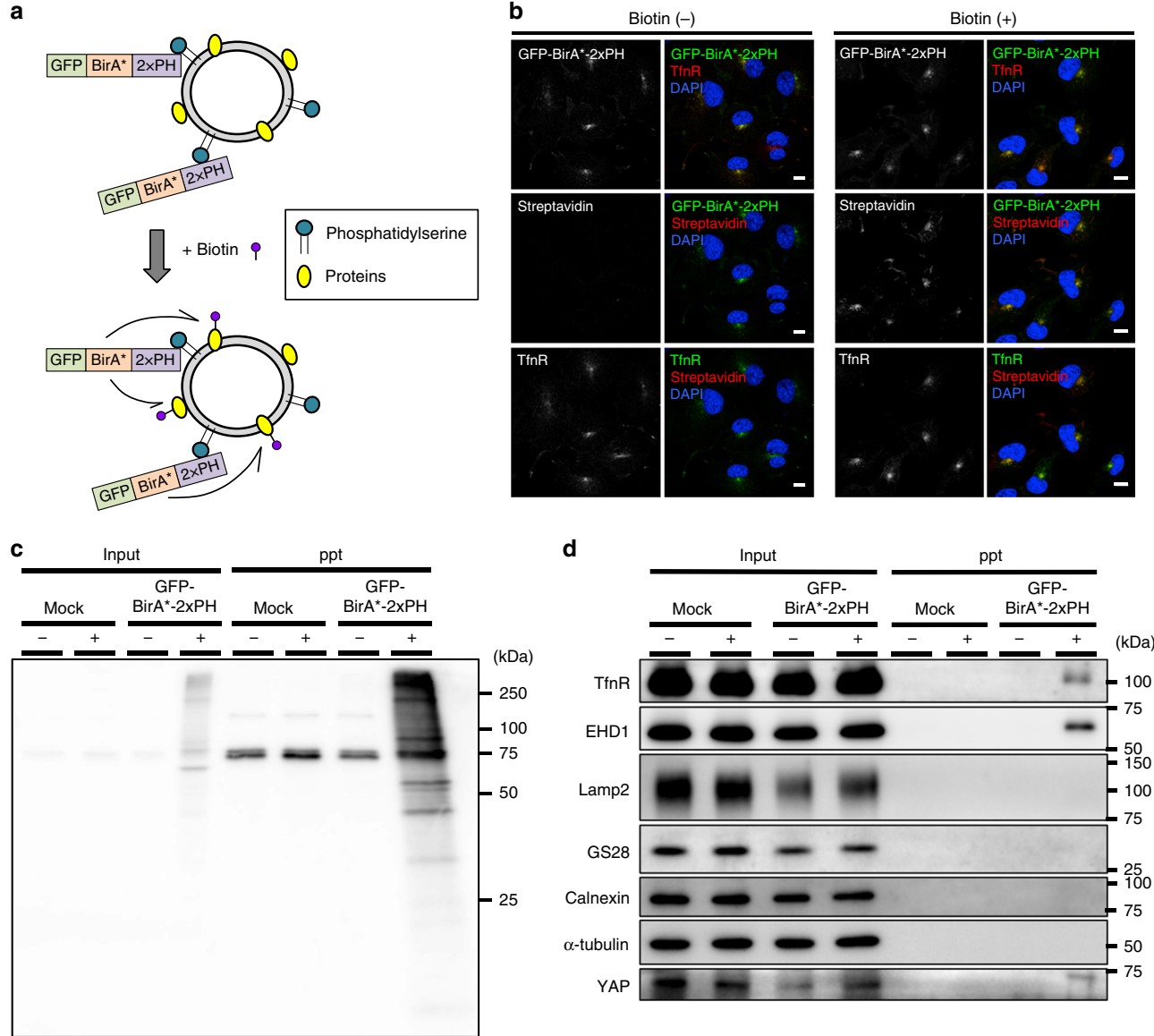

**Fig. 1** Identification of proteins proximal to PS in live cells. **a** Schematic illustration of biotinylation of proteins proximal to PS with GFP-BirA*-2xPH. **b** COS-1 cells stably expressing GFP-BirA*-2xPH were incubated with or without 50 μM biotin for 24 h. The cells were then fixed, permeabilized, and stained for TfnR and biotin with Alexa594-streptavidin. **c**, **d** Lysates from cells in **b** were mixed with streptavidin-coupled magnetic beads. The proteins pulled down by the beads were blotted with streptavidin-HRP in **c**, or the indicated antibodies in **d**. Nuclei were stained with DAPI. Scale bars, 10 μm

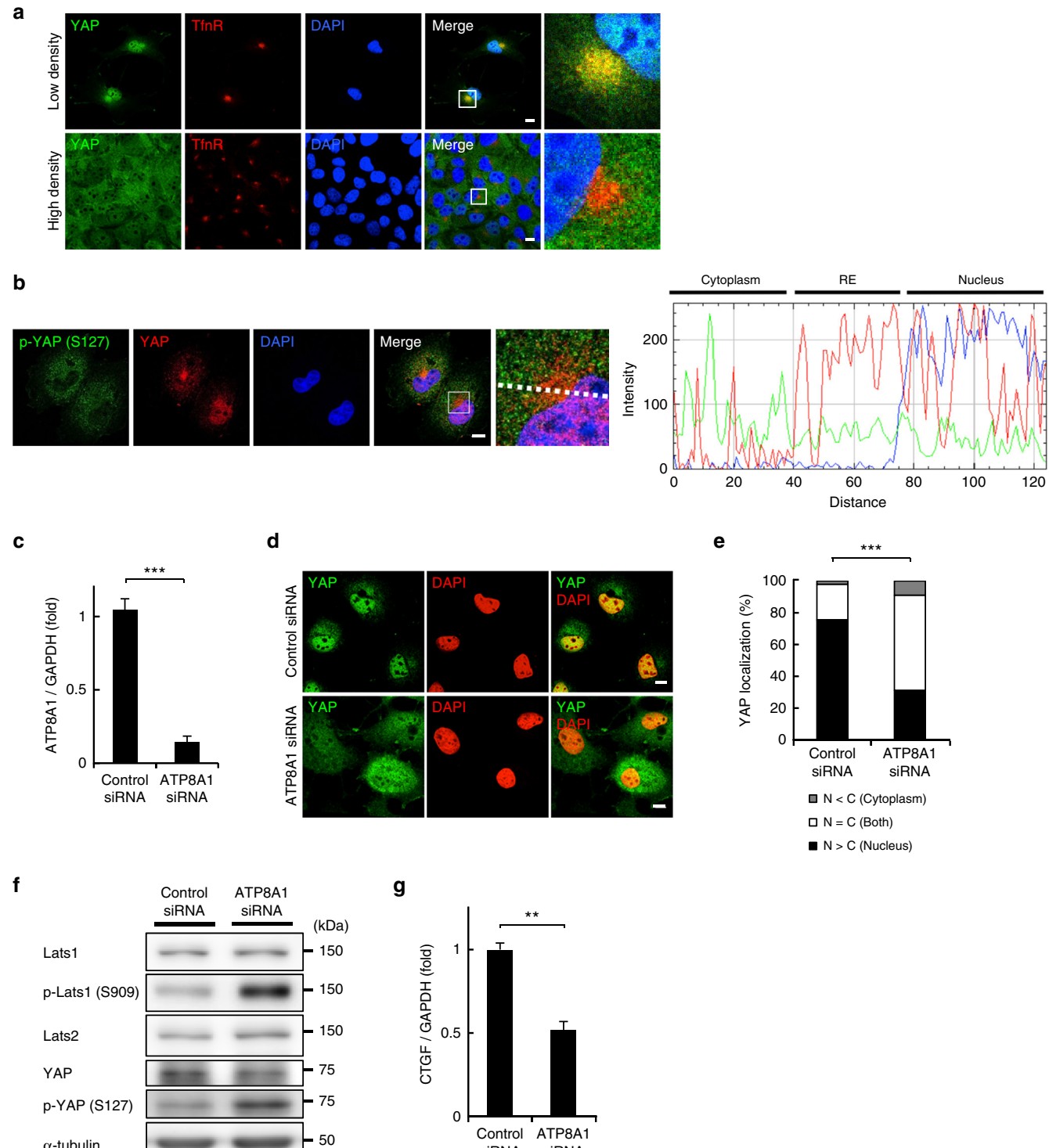

**Fig. 2** A PS-flippase at REs contributes to the nuclear localization of YAP. **a** COS-1 cells at low density or high density were fixed, permeabilized, and stained for YAP and TfnR. Magnified images of the boxed areas around the perinuclear REs are shown in the right column. **b** COS-1 cells at low density were fixed, permeabilized, and stained for YAP and phosphorylated YAP (S127). A fluorescence intensity profile along the dotted line in the magnified image is shown in the right panel. **c** qRT-PCR analysis of ATP8A1 mRNA in COS-1 cells treated with ATP8A1 siRNA for 48 h. GAPDH was used as an internal control. **d** COS-1 cells were treated with control or ATP8A1 siRNA for 48 h. The cells were then fixed, permeabilized, and stained for YAP. **e** Subcellular localization of YAP in cells in **d** was examined. **f** Lysates from cells in **d** were immunoblotted for the indicated proteins. α-tubulin was used as a loading control. **g** qRT-PCR analysis of CTGF mRNA from cells in **d**. GAPDH was used as an internal control. Data are mean ± s.d. from two (for **e**, $n > 40$ cells) or three (for **c**, **g**) independent experiments. Statistical significance was determined with two-tailed Student's $t$-test (for **c**, **g**) or two-sided Fisher's exact test (for **e**); **$P <$ 0.01, ***$P < 0.001$. Nuclei were stained with DAPI. Scale bars, 10 μm

expected, GFP-BirA*-2xPH, like 2xPH, co-localized with an RE protein transferrin receptor (TfnR) (Fig. 1b and Supplementary Fig. 1). Biotin was then added to the culture medium to biotinylate proteins proximal to GFP-BirA*-2xPH in living cells. Biotinylated proteins were detected with the fluorescent probe Alexa-streptavidin. As expected, Alexa-streptavidin mostly co-localized with TfnR (Fig. 1b). Western blots showed that TfnR and another RE protein, EHD1[16, 17] were biotinylated, while several proteins at other subcellular sites (Lamp2 in lysosomes, GS28 in the Golgi, calnexin in the ER, and α-tubulin in the cytosol) were not (Fig. 1c, d). Together, these findings indicated that RE proteins were preferentially biotinylated.

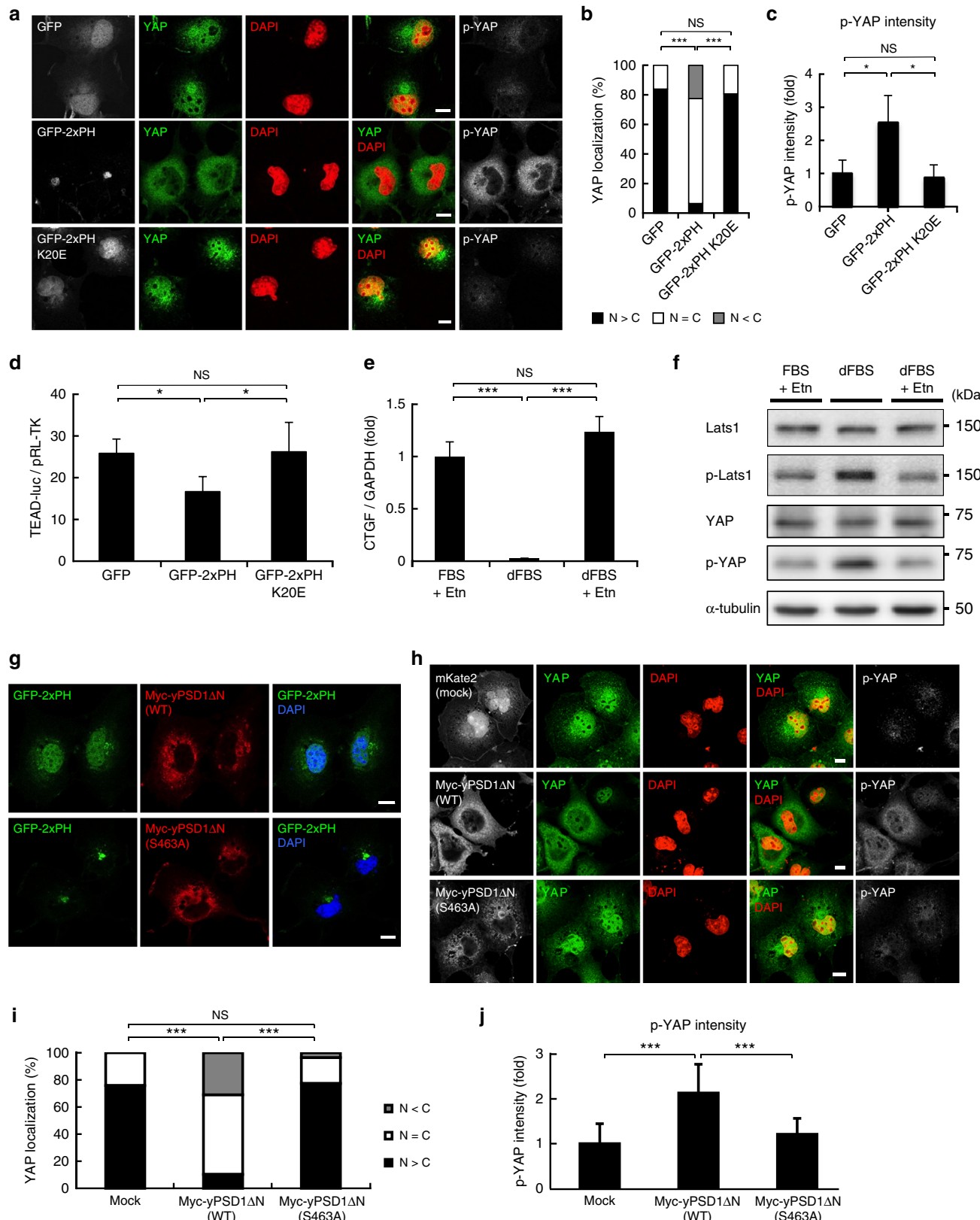

We then isolated the biotinylated proteins with streptavidin-coated magnetic beads and analyzed them by mass spectrometry. About 400 biotinylated proteins were identified (Supplementary Data 1). Of these, 113 proteins are reported to be associated with endosomes (Supplementary Data 1). In addition to EHD1, several RE proteins that function in membrane trafficking were identified, including VAMP3[18], Rab11-FIP1[19], MICAL-L1[20], and SMAP2[21].

Interestingly, we found that YAP and a group of proteins associated with the YAP signalling pathway were biotinylated (Supplementary Table 1). Western blot confirmed the presence of YAP in the streptavidin-bead fraction (Fig. 1d). In low-density (i.e., proliferating) cells, YAP regulates target genes that are essential for cell proliferation in the nucleus[22, 23]. Although YAP has no apparent membrane-associated domain, our results suggested that YAP was at least partly associated with the RE membranes. Indeed, immunofluorescence imaging showed that YAP localized at REs in low-density cells, in addition to its expected localization in the nucleus (Fig. 2a and Supplementary Fig. 2a). YAP was also observed at REs in low-density HEK293A cells (Supplementary Fig. 2b). When the cells became confluent (i.e., stopped proliferating), the RE localization of YAP was abolished (Fig. 2a and Supplementary Fig. 2b).

YAP can be phosphorylated on Ser127 by Lats kinases (Lats1 and Lats2)[22]. Phosphorylated YAP binds to 14-3-3 proteins in the cytoplasm, which prevents the translocation of YAP to the nucleus[22, 24]. Although YAP was found to be enriched in REs in low-density cells (Fig. 2a), phosphorylated YAP was evenly distributed throughout the cytoplasm (Fig. 2b). The result suggested that YAP at REs was preferentially in an unphosphorylated form (Fig. 2b). As expected, phosphorylated YAP was excluded from the nucleus.

**PS has a role in YAP activation**. ATP8A1 is a PS-flippase that localizes to REs. As a flippase, it catalyzes the enrichment of PS in the cytoplasmic leaflet of RE membranes[9]. We found that the knockdown of ATP8A1 with siRNA (1) significantly reduced the nuclear localization of YAP (Fig. 2c–e), (2) increased the phosphorylation of YAP (Fig. 2f), (3) increased the phosphorylation of Lats1 on Ser909, an activated form of Lats1[25] (Fig. 2f), and (4) significantly decreased the mRNA expression of CTGF, a YAP-regulated gene (Fig. 2g). The increase of phosphorylated YAP was reduced by depletion of Lats1/2 (Supplementary Fig. 3). These results indicated that the concentrated PS in the cytoplasmic leaflet of RE membranes is critical for the nuclear localization of YAP and the transcription of YAP-downstream genes by suppressing Lats1 activity.

We examined the involvement of PS in YAP signalling by other approaches. First, the PS-specific probe GFP-2xPH[9] was used to mask the PS at the cytosolic leaflet of RE membranes. Overexpression of GFP-2xPH in the cytoplasm significantly reduced the nuclear localization of YAP (Fig. 3a, b), increased the level of phosphorylated YAP (Fig. 3c), and reduced the transcriptional activity of YAP (Fig. 3d) using the TEAD reporter system[26]. In contrast, overexpression of the mutant (GFP-2xPH K20E)[9], which does not bind PS, had no effect (Fig. 3a–d). Second, YAP signalling was examined in PS-auxotrophic mutant CHO cells (PSA-3)[27] that lack the activity of phosphatidylserine synthase-1. When PSA-3 cells are cultured with dialyzed fetal bovine serum (dFBS) that lacks ethanolamine (Etn), cellular levels of PS decrease by around 30% compared with PSA-3 cells cultured with dFBS plus Etn[28]. The PS-deficient culture conditions significantly reduced the mRNA expression of CTGF (Fig. 3e) and increased the level of phosphorylated YAP and phosphorylated Lats1 (Fig. 3f). In contrast, including Etn in the culture medium to maintain cellular levels of PS[28] did not affect the mRNA expression of CTGF, or the level of phosphorylated YAP or phosphorylated Lats1 (Fig. 3e, f). Third, we made use of overexpression of phosphatidylserine decarboxylase 1 (PSD1), a mitochondrial enzyme that degrades PS[29–31]. We generated a yeast PSD1 mutant (yPSD1ΔN) that lacks an N-terminal mitochondria-targeting sequence and a transmembrane domain. The RE localization of the PS probe 2xPH was drastically lost in cells that overexpress yPSD1ΔN, but not catalytically inactive yPSD1ΔN (S463A) (Fig. 3g), indicating that overexpression of yPSD1ΔN reduced PS in REs. We then examined the effect of yPSD1ΔN overexpression on YAP. The expression of yPSD1ΔN, but not yPSD1ΔN (S463A), suppressed the nuclear translocation of YAP (Fig. 3h, i) and increased the level of phosphorylated YAP (Fig. 3j). These results provide further evidence that PS has a role in YAP activation.

**Evectin-2 contributes to YAP activation**. Evectin-2 is an RE-resident protein that regulates the retrograde trafficking from REs to the Golgi[7]. Intriguingly, depletion of evectin-2 by siRNA significantly reduced the nuclear localization of YAP and mRNA expression of CTGF in low-density COS-1 cells (Fig. 4a–c and Supplementary Fig. 4a). In contrast to the knockdown of evectin-2, knockdown of other proteins essential for RE membrane traffic, such as a small GTPase Rab11 or a v-SNARE VAMP3[32], did not affect the nuclear localization of YAP (Fig. 4d, e and Supplementary Fig. 4b, c). These results suggest that evectin-2 functions in YAP signalling independently of the regulation of RE membrane traffic.

Dysregulation of the YAP signalling pathway has been linked to tumorigenesis in several cancers[33, 34]. MDA-MB-231 cells are aggressive metastatic breast cancer cells, whose proliferation depends on YAP signalling[35]. Knockdown of evectin-2 in MDA-MB-231 cells significantly reduced mRNA expression of CTGF (Fig. 4f), increased the level of Lats1 and phosphorylated YAP

**Fig. 3** Cellular PS levels contribute to the nuclear localization of YAP. **a** GFP, GFP-2xPH (evectin-2, WT), or GFP-2xPH (K20E) was expressed in COS-1 cells for 24 h. The cells were then fixed, permeabilized, and stained for YAP and p-YAP. **b** Subcellular localization of YAP in cells in **a** was examined. **c** The fluorescence intensity of p-YAP in **a** was quantified and normalized to that of GFP-transfected cells. **d** Transcriptional coactivity of YAP was examined by the TEAD reporter system. GFP, GFP-2xPH, or GFP-2xPH (K20E) was co-expressed with 8xGTIIC TEAD reporter and pRL-TK (as internal control). **e** PSA-3 cells were cultured for 72 h with three different conditions (FBS + 10 μM Etn, dFBS, or dFBS + 40 μM Etn). qRT-PCR analysis of CTGF mRNA in these cells was then performed. GAPDH was used as an internal control. **f** Lysates from cells in **e** were immunoblotted for the indicated proteins. α-tubulin was used as a loading control. **g** COS-1 cells were cotransfected with GFP-2xPH and Myc-yPSD1ΔN (WT or S463A). After 24 h, the cells were fixed and stained for Myc. **h** COS-1 cells were transfected with mKate2, Myc-yPSD1ΔN (WT), or Myc-yPSD1ΔN (S463A). After 24 h, the cells were fixed and costained for YAP, p-YAP, and Myc. **i** Subcellular localization of YAP in cells in **h** was examined. **j** The fluorescence intensity of p-YAP in **h** was quantified and normalized to that of mKate2-transfected cells. Data are mean ± s.d. from two (for **b**, **c**, $n > 30$ cells), three (for **e**), three (for **i**, **j**, $n > 85$ cells), or six (for **d**) independent experiments. Statistical significance was determined with one-way analysis of variance followed by Tukey–Kramer post hoc test (for **c**, **d**, **e**, **j**) or Kruskal–Wallis test followed by Steel–Dwass post hoc test (for **b**, **i**); *$P < 0.05$, ***$P < 0.001$, NS not significant. The nuclei were stained with DAPI. Scale bars, 10 μm

(Fig. 4g), and reduced the cell proliferation rate (Fig. 4h and Supplementary Fig. 4d). Further knockdown of Lats proteins significantly restored the cell proliferation rate (Fig. 4i), suggesting that inhibition of proliferation by evectin-2 knockdown was mediated by YAP inhibition. Knockdown of ATP8A1 also significantly reduced the cell proliferation rate (Fig. 4h).

**Evectin-2 activates Nedd4 E3 ligases that degrade Lats1.** To elucidate the mechanism by which evectin-2 regulates the amount of Lats1, we focused on evectin-2-binding proteins. From immunoprecipitates of FLAG-tagged evectin-2, we identified five peptides from three proteins (Itch, WWP1, and WWP2), all of which are Nedd4 ubiquitin E3 ligases (Supplementary Fig. 5a, b). Indeed, FLAG-evectin-2 expressed in COS-1 cells immunoprecipitated endogenous Itch, WWP1, and WWP2 (Fig. 5a). Immunofluorescence imaging showed that these E3 ligases colocalized with evectin-2 at REs (Fig. 5b and Supplementary Fig. 5c). In breast cancer cells, Itch and WWP1 ubiquitinate Lats1[36, 37], which leads to an increase of non-phosphorylated

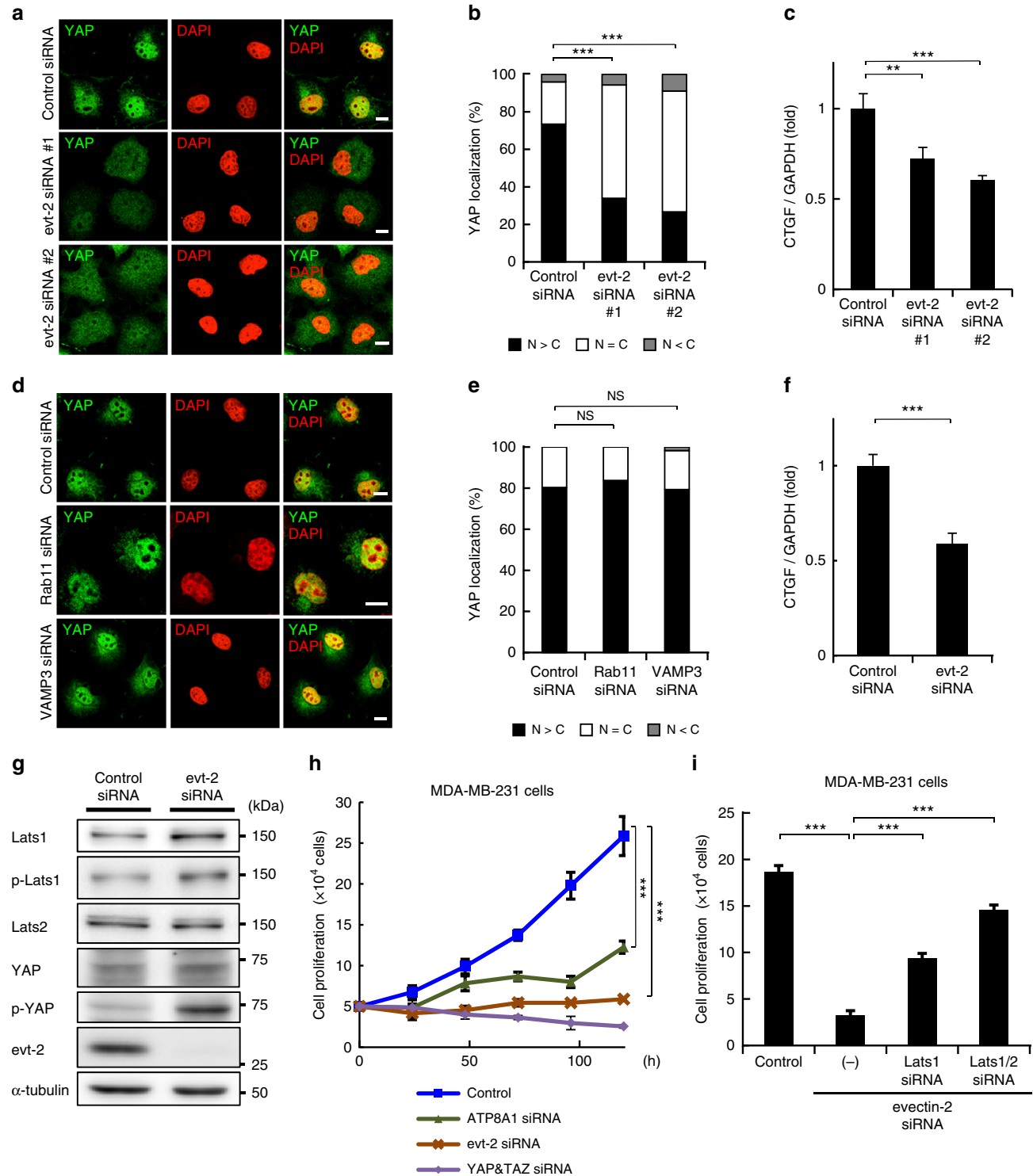

YAP[36]. Knockdown of WWP1 or WWP2 in COS-1 cells decreased the nuclear localization of YAP (Fig. 5c, d) and mRNA expression of CTGF (Fig. 5e) and increased the levels of Lats1 and phosphorylated YAP (Fig. 5f). Double knockdown of WWP1 and WWP2 did not have an additive effect on the levels of Lats1, phosphorylated YAP, or CTGF. Knockdown of Itch did not cause these changes in COS-1 cells.

We examined whether evectin-2 contributes to the stability of Lats1. The cells were treated with cycloheximide (CHX), an agent that inhibits protein translation, and the level of Lats1 was monitored. In control cells, the amount of Lats1 was decreased by 40% after 12 h treatment with CHX (Supplementary Fig. 6a). In contrast, in evectin-2-knockdown cells, the amount of Lats1 was decreased by only 15%. These results indicated that the stability of Lats1 was negatively regulated by evectin-2. We then examined the ubiquitination of Lats1. In control cells, after 4 and 8 h treatment with the proteasome inhibitor MG132, ubiquitination of Lats1 was detected (Supplementary Fig. 6b). Importantly, the ubiquitination of Lats1 was mostly abolished by evectin-2 knockdown. These results suggested that evectin-2 negatively regulates the stability of Lats1 through the ubiquitination of Lats1. Knockdown of ATP8A1, which did not affect the amount of Lats1 (Fig. 2f), did not affect the stability of Lats1 (Supplementary Fig. 6a) or the ubiquitination of Lats1 (Supplementary Fig. 6b).

Nedd4 E3 ligases have multiple WW domains (Supplementary Fig. 5b). WW domains bind several proline-rich motifs, including the PPXY motif, where X is any amino acid[38, 39]. Curiously, evectin-2 has a PPPY sequence starting at Pro129 (Supplementary Fig. 5d). Evectin-2 mutants that lacked the PPPY sequence or had a point mutation at Tyr132 did not bind to Itch, WWP1, or WWP2 (Fig. 5g). More importantly, the nuclear localization of YAP in evectin-2-depleted cells was rescued by the expression of siRNA-resistant wild-type (WT) evectin-2, but not by the expression of an siRNA-resistant PPPY mutant of evectin-2 (Fig. 5h, i). These results suggest that binding of the Nedd4 E3 ligases to the PPPY sequence of evectin-2 is critical for the nuclear translocation of YAP. In the case of Itch, intramolecular interactions inhibit its catalytic activity, and once the inhibition is released, it is activated and self-ubiquitinates[40]. We found that FLAG-tagged Itch, WWP1, and WWP2 were ubiquitinated in cells that co-express evectin-2 (WT), but not the PPPY mutant of evectin-2 (Fig. 5j and Supplementary Fig. 7a, b). These results suggest that the binding of evectin-2 to Nedd4 E3 ligases increases their catalytic activity. Knockdown of evectin-2 did not alter the RE localizations of Itch, WWP1, and WWP2 (Supplementary Fig. 8a), implying that evectin-2 does not appear to have a large role in the recruitment of the Nedd4 E3 ligases to REs. The observation that the Nedd4 E3 ligases that lack the N-terminal C2 domain did not localize to REs (Supplementary Fig. 8b) suggested that the C2 domain has a role in the recruitment of the Nedd4 E3 ligases to REs.

## Discussion

Our analysis of proteins proximal to PS suggested that RE membranes have a role in YAP activation and cell proliferation in low-density cells (Fig. 5k). Knockdown of ATP8A1 (an RE PS-flippase) or evectin-2 (an RE-resident protein) inhibited the nuclear localization and signalling of YAP. However, the mechanisms of YAP activation by ATP8A1 and evectin-2 appeared to be different. ATP8A1 knockdown increased the phosphorylated (activated) form of Lats1, but did not affect the amount of Lats1 or its ubiquitination. Evectin-2 knockdown reduced the ubiquitination and increased the level of Lats1.

In ATP8A1-knockdown cells, PS was still present in the cytosolic leaflet of RE membranes, and evectin-2 localized normally at REs[9]. Therefore, ATP8A1 knockdown may not affect evectin-2 and would not influence the ubiquitination or amounts of Lats1. It is not clear how the concentrated PS in the cytosolic leaflet of RE membranes, which is generated by ATP8A1, functions in reducing the phosphorylation of Lats1. We found that evectin-2 enhanced the ubiquitination and degradation of Lats1 through the binding to Nedd4 E3 ligases. Although evectin-2 is a PS-binding protein[7], it is not clear at present whether PS is involved in the evectin-2-mediated Lats1 degradation. Further studies are needed to elucidate the precise molecular mechanism by which PS in the REs activates YAP.

Of clinical relevance is our finding that the PS-flippase and evectin-2/Nedd4 E3 ligase were active in an aggressive metastatic breast cancer cell line, which could lead to new drug targets for cancers in which the malignancy depends on YAP function. Finally, the BioID method may be useful for revealing the functions of other membrane phospholipids through the identification of proteins proximal to a specific phospholipid.

## Methods

**Plasmids.** For the experiment in Fig. 5j, Myc-tagged evectin-2 constructs were used[7]. pcDNA3.1 Myc-BirA* (#35700) was purchased from Addgene. GFP-BirA*-2xPH was generated from pEGFP-C2 evectin-2 2xPH and Myc-BirA* using 5′-GCGCGAATTCAATGGAACAAAAACTCATCTCAG-3′ (sense primer, EcoRI site is underlined) and 5′-TTAAGCTTGGTACCGAGCTCGCGGCCGCCTAGTT-3′ (antisense primer, NotI site is underlined). FLAG-tagged evectin-2 ΔPPPY mutant was generated from FLAG-tagged evectin-2 (WT) using 5′-ACGGCCTATGCTGCACCGGCCCCTGAG-3′ (sense primer) and 5′-TGAGGGAAACCACGGATGTCTCATCGGTCATG-3′ (antisense primer). FLAG-tagged evectin-2 PPPA mutant was generated from FLAG-tagged evectin-2 (WT) using 5′-CACCTCCACCAGCCACGGCCTATGC-3′ (sense primer) and 5′-AGGAAACCACGGATGTCTCATCGGT-3′ (antisense primer). GFP-2xPH K20E mutant was generated from pEGFP-C2 evectin-2 2xPH and Myc-evectin-2 K20E using In-Fusion HD Cloning kit (TAKARA) and following primers: 5′-GGCCGGACTCAGATCTCGATGGCGTTTGTGAAGAGTGG-3′ (sense primer 1, BglII site is underlined); 5′-CATGGTACCGTCGACGTTTGTCCTAGAATCTTGGA-3′ (antisense primer 1); 5′-GTCGACGGTACCATGGCGTTTGTGAAGAGTGGCTG-3′ (sense primer 2); and 5′-TAGATCCGGTGGATCCCTAGTTTGTCCTAGAATCTTGGAG-3′ (antisense primer 2, BamHI site is underlined). yPSD1ΔN was amplified by PCR with synthesized DNA of codon-optimized yeast PSD1 (gBlocks Gene Fragments, Integrated DNA Technologies) using the following primers: 5′-GCTTGGATCCGAATTCGGTGGCGGTGGCTCGGGCGGTGGTGGGTCGGGTGGCGGCGGATCTGACAGCACCGAGGAGGACGC-3′ (sense primer, EcoRI site

**Fig. 4** Evectin-2 contributes to the nuclear localization of YAP. **a** COS-1 cells were treated with control siRNA, evectin-2 siRNA#1, or evectin-2 siRNA#2 for 48 h. The cells were then fixed, permeabilized, and stained for YAP. **b** Subcellular localization of YAP in cells in **a** was examined. **c** qRT-PCR analysis of CTGF mRNA from cells in **a**. GAPDH was used as an internal control. **d** COS-1 cells were treated with control, Rab11, or VAMP3 siRNA for 48 h. The cells were then fixed, permeabilized, and stained for YAP. **e** Subcellular localization of YAP in cells in **d** was examined. **f** qRT-PCR analysis of CTGF mRNA in MDA-MB-231 cells depleted of evectin-2. GAPDH was used as an internal control. **g** MDA-MB-231 cells were treated with evectin-2 siRNA for 72 h. Cells were lysed and immunoblotted for the indicated proteins. α-tubulin was used as a loading control. **h** MDA-MB-231 cells were treated with siRNA for evectin-2, ATP8A1, or YAP/TAZ for 48 h and then trypsinized. A total of $5 \times 10^4$ cells were replated and further treated with the same siRNA 24 h after replating. The cells were counted every 24 h after replating. **i** MDA-MB-231 cells were treated with siRNA for evectin-2, evectin-2/Lats1, or evectin-2/Lats1/Lats2 for 48 h and then trypsinized. A total of $5 \times 10^4$ cells were replated and further treated with the same siRNA 24 h after replating. The cells were counted 96 h after replating. Data are mean ± s.d. from two (for **b**, **e**, $n > 40$ cells) or three (for **c**, **f**, **h**, **i**) independent experiments. Statistical significance was determined with two-tailed Student's $t$ test (for **f**), one-way analysis of variance followed by Tukey–Kramer post hoc test (for **c**, **h**, **i**), or Kruskal–Wallis test followed by Steel–Dwass post hoc test (for **b**, **e**); **$P < 0.01$, ***$P < 0.001$, NS not significant. The nuclei were stained with DAPI. Scale bars, 10 μm

is underlined) and 5′-CTTTGTAGTCGTCGACTCACTTCAGGTCGTTCTTGC-3′ (antisense primer, SalI site is underlined). The product encoding yeast PSD1ΔN (aa. 99-500) was introduced into pcDNA3 to generate N-terminal myc-tagged construct. The corresponding catalytically inactive yeast PSD1ΔN (S463A) was generated by site-directed mutagenesis. 8xGTIIC TEAD luciferase reporter vector (#34615) was purchased from Addgene. For the preparation of cumate-inducible siRNA-resistant Myc evectin-2 WT or PPPA mutant, six silent mutations were

introduced by PCR in the target sequence of evectin-2 siRNA#2 (see below) using 5′-TATCCCTACGCCGGACTTTATGGACAGCAGCC-3′ (sense primer) and 5′-CTGATAAGGCACGGCATACGCCTGCCC-3′ (antisense primer). The siRNA-resistant evectin-2 constructs were then introduced into cumate-inducible enhanced episomal vector (EEV #610-A1) (System Biosciences). Human Itch, WWP1, and WWP2 constructs[41] were kindly provided by Dr Wesley I. Sundquist (University of Utah, US). N-terminal GFP-tagged Itch, WWP1, and WWP2 were

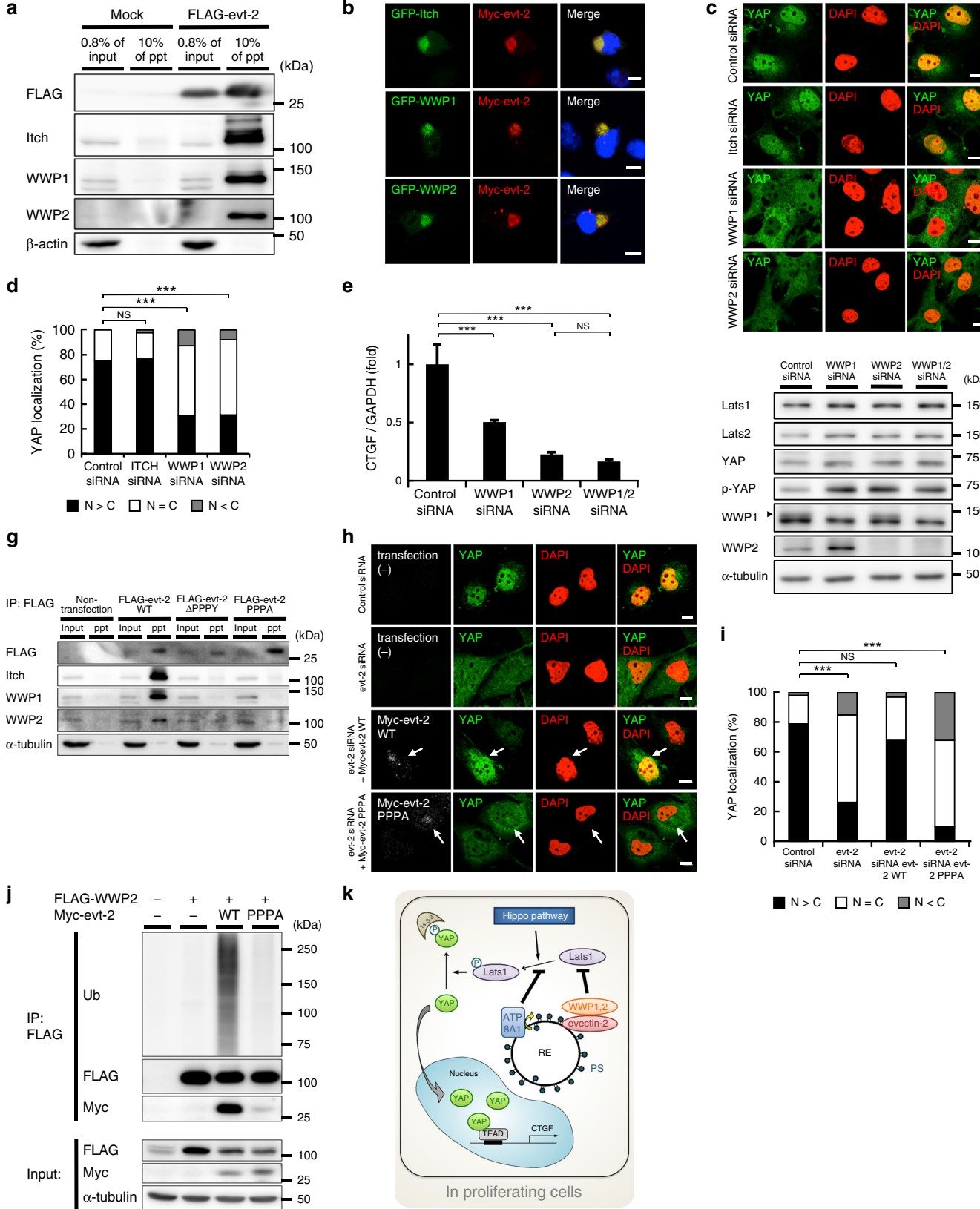

generated in pMXs-IP vector. FLAG-tagged ItchΔC2 mutant was generated from FLAG-Itch (WT) using 5′-GGGCTACAGTTAGAGTCTGAAGTTGTTACCAA TGGTGAAACT-3′ (sense primer) and 5′-CATGGTGAGGCTACCCATTGAACC AAGTTGTGATCC-3′ (antisense primer). FLAG-tagged WWP1ΔC2 mutant was generated from FLAG-WWP1 (WT) using 5′-GGATTGGTGATTGAGCAAGAA AATATAACAAACTGCAGCT-3′ (sense primer) and 5′-TCCACTGTGGTTATT ACTAGTATCAGACCTTGGTGA-3′ (antisense primer). FLAG-tagged WWP2ΔC2 mutant was generated from FLAG-WWP2 (WT) using 5′-GATGGAT CACAGCTGCCTTCGAGAGACTCCAGTGGA-3′ (sense primer) and 5′-CTCAA AAGGCAGGGCCACTCCTGCCCG-3′ (antisense primer). GFP-Lats1 (#19053) was purchased from Addgene.

**Antibodies**. Antibodies used in this study were as follows: rabbit anti-WWP1 (ab43791, dilution 1:1000), mouse anti-β-actin (ab8226, dilution 1:1000), and rabbit anti-EHD1 (ab109747, dilution 1:1000) (Abcam); mouse anti-α-tubulin (T-9026, dilution 1:4000) and mouse anti-Myc (9E10, dilution 1:500) (Sigma); rabbit anti-p-YAP (S127) (#13008, dilution 1:1000 for western blot, #4911, dilution 1:250 for immunofluorescence), rabbit anti-p-Lats1 (S909) (#9157, dilution 1:1000), and rabbit anti-FLAG (#2368, dilution 1:1000 for immunofluorescence and western blotting) (Cell Signaling Technology); mouse anti-TfnR (H68.4, dilution 1:1000 for immunofluorescence and western blotting) and rabbit anti-Rab11 (71-5300, dilution 1:1000 for immunofluorescence and western blotting) (Thermo); rabbit anti-syntaxin 5 (110053, dilution 1:1000, Synaptic Systems); sheep anti-TGN46 (AHP500G, dilution 1:4000, Serotec); goat anti-VPS26A (EB06256, dilution 1:200, Everest Biotech); mouse anti-Lamp2 (sc-18822, dilution 1:200) and rabbit anti-WWP2 (sc-30052, dilution 1:500) (Santa Cruz); rabbit anti-Myc (#06-549, dilution 1:1000, Millipore); rabbit anti-Lats1 (A300-477A, dilution 1:4000, Bethyl Laboratories); sheep anti-GM130 (AF8199, dilution 1:1000, R&D Systems); mouse anti-calnexin (610523, dilution 1:1000 for immunofluorescence and western blotting), mouse anti-EEA1 (610456, dilution 1:1000), mouse anti-GS28 antibody (611184, dilution 1:1000), mouse anti-GM130 (610823, dilution 1:1000), and mouse anti-Itch (611198, dilution 1:1000) (BD Transduction Laboratories); anti-mouse CD63 (RFAC4, dilution 1:1000, Cymbus Biotechnology LTD); mouse anti-KDEL (ADI-SPA-827, dilution 1:1000, Enzo Life Sciences); mouse anti-Ubiquitin (sc-8017, dilution 1:1000, Santa Cruz); sheep anti-mouse-IgG conjugated to horseradish peroxidase (HRP) and donkey anti-rabbit-IgG conjugated to HRP (dilution 1:2000, GE Healthcare); goat anti-rat-IgG conjugated to HRP (dilution 1:2000, American Qalex Antibodies); Pierce™ High Sensitivity Streptavidin-HRP (dilution 1:4000, Thermo); and Alexa-Fluor-conjugated secondary antibodies (dilution 1:2000, Invitrogen). For immunofluorescence detection and western blotting of YAP, rat monoclonal anti-YAP antibody[42] was used (dilution 1:1000). For western blotting of evectin-2, rat monoclonal anti-evectin-2 antibody[7] was used (dilution 1:200).

**Reagents**. The following reagents were purchased from the manufacturers as noted: cumate (System Biosciences); biotin and Dynabeads® MyOne streptavidin C1 (Thermo); ethanolamine and anti-FLAG M2 Affinity Gel (Sigma); and CHX (Wako).

**Cell culture**. COS-1, HEK293A, and HEK293T cells (American Type Culture Collection) were cultured at 37 °C with 5% CO$_2$ in DMEM (Sigma) containing 10% heat-inactivated fetal bovine serum (FBS)/penicillin/streptomycin/glutamine (PSG). PSA-3 cells were maintained in Ham's F-12 medium (Wako Pure Chemicals) supplemented with 10% heat-inactivated FBS/PSG/10 μM Etn. MDA-MB-231 cells (American Type Culture Collection) were cultured at 37 °C without CO$_2$ in Leibovitz's L-15 medium (Wako) containing 10% heat-inactivated FBS/PSG.

COS-1 cells that stably express GFP-BirA*-2xPH, GFP-Itch, GFP-WWP1, or GFP-WWP2 were established using retrovirus transfection: HEK293T cells were transfected with pMXs-IP-EGFP-BirA*-2xPH, Itch, WWP1, or WWP2 together with pCG-VSV-G, and the medium that contains the retrovirus was collected. COS-1 cells were incubated with the medium and then selected with puromycin for a week.

COS-1 cells that express siRNA-resistant Myc-evectin-2 in a cumate-dependent fashion were established by selection with puromycin for a week.

**Plasmid transfection**. The cells were transiently transfected with plasmids using Lipofectamine 2000 (Invitrogen) according to the manufacturer's instructions.

**RNA interference**. siRNA specific to WWP1 (Silencer® Select siRNA, s21788) and TAZ (Silencer Select® siRNA, s24787) were purchased from Thermo. siRNA specific to YAP (siGENOME SMARTpool, M-012200-00) and VAMP3 (siGENOME SMARTpool, M-011934-01) were purchased from Dharmacon. Negative control siRNA was purchased from Nippon Gene. siRNA duplex oligomers were designed (Nippon Gene) as follows: CUCAAAUGUGGAACGGAUU (ATP8A1 siRNA), CUGCAUGCUCCAGAUUGUU (evectin-2 siRNA#1), CUACCAGUACCCAUAU GCA (evectin-2 siRNA#2), CGGAUGGUCACCUGAUCUA (evectin-2 siRNA#3, for MDA-MB-231 cells), CAAGAUCUGCGAAGACGUU (ITCH siRNA), GUCUCGAUUUACUCGAAAU (Rab11 siRNA), CAGGAUGGGAGAUGAAA UA (WWP2 siRNA), AUUCGGGAAUCCCUUAGGA (Lats1 siRNA for COS-1 cells), CACGGCAAGAUAGCAUGGA (Lats1 siRNA for MDA-MB-231 cells), AAAGGCGUAUGGCGAGUAG (Lats2 siRNA for COS-1 cells), and GCCACGAC UUAUUCUGGAA (Lats2 siRNA for MDA-MB-231 cells). A total of 10–20 nM siRNA was introduced to cells using Lipofectamine RNAiMAX (Invitrogen) according to the manufacturer's instruction. After 4 h, the medium was replaced by DMEM or L-15 with 10% heat-inactivated FBS/2 mM L-glutamine and cells were further incubated for 44 or 68 h for subsequent experiments.

**Affinity capture of biotinylated proteins**. The BioID with GFP-BirA*-2xPH was performed as follows: COS-1 cells that stably express GFP-BirA*-2xPH were incubated for 24 h in complete media supplemented with or without 50 μM biotin. After three PBS washes, the cells (~1 × 10$^7$) were scraped in HEPES Buffer (20 mM HEPES-NaOH pH 7.3, 0.25 M sucrose, 1 mM EDTA pH 8.0, and cOmplete EDTA-free Protease inhibitor cocktail (Roche)) and pelleted by centrifugation at 2000 rpm for 5 min at 4 °C. The cells were lysed at 25 °C in 1 ml lysis buffer (50 mM Tris, pH 7.4, 500 mM NaCl, 0.4% SDS, 5 mM EDTA, 1 mM DTT, and cOmplete EDTA-free Protease inhibitor cocktail (Roche)) with sonication. Triton X-100 was added to 2% final concentration. After further sonication, an equal volume of 50 mM Tris-HCl (pH 7.4) was added before additional sonication (subsequent steps at 4 °C) and centrifugation at 13,500 rpm. Supernatants were incubated with 120 μl Dynabeads (MyOne Streptavidin C1 (Thermo)) overnight. The beads were collected and washed twice for 8 min at 25 °C (all subsequent steps at 25 °C) in 1 ml wash buffer 1 (2% SDS in H$_2$O). This was repeated once with wash buffer 2 (0.1% deoxycholate, 1% Triton X-100, 500 mM NaCl, 1 mM EDTA, and 50 mM HEPES, pH 7.5), once with wash buffer 3 (250 mM LiCl, 0.5% NP-40, 0.5% deoxycholate, 1 mM EDTA, and 10 mM Tris, pH 8.1), and twice with wash buffer 4 (50 mM Tris, pH 7.4, and 50 mM NaCl). Bound proteins were removed from the magnetic beads with 50 μl of SDS-sample buffer (final 62.5 mM Tris-HCl, pH 6.8, 10% glycerol, 1% SDS, 100 mM DTT, and 0.005% BPB) saturated with 3 mM biotin at 98 °C for 5 min.

**Fig. 5** Binding of evectin-2 and Nedd4 E3 ligases is required for the nuclear localization of YAP. **a** FLAG-tagged evectin-2 was expressed in COS-1 cells for 24 h. Cell lysates with 1% Triton X-100 were immunoprecipitated with anti-FLAG antibody. The lysate (0.8%) and the immunoprecipitated fraction (10%) were then blotted for FLAG, Itch, WWP1, or WWP2. β-actin was used as a loading control. **b** Indicated proteins were expressed in COS-1 cells. The cells were then fixed, permeabilized, and stained for Myc. **c** COS-1 cells were treated with control, Itch, WWP1, or WWP2 siRNA for 48 h. The cells were then fixed, permeabilized, and stained for YAP. **d** Subcellular localization of YAP in cells in **c** was examined. **e** COS-1 cells were treated with the indicated siRNAs for 48 h. qRT-PCR analysis of CTGF mRNA was performed. GAPDH was used as an internal control. **f** The lysate of COS-1 cells in **e** was immunoblotted for the indicated proteins. α-tubulin was used as a loading control. A closed arrowhead indicates WWP1. **g** FLAG-tagged evectin-2 protein (WT or mutants) was expressed in COS-1 cells for 24 h. Cell lysates with 1% Triton X-100 were immunoprecipitated with anti-FLAG antibody. The lysate (0.8%) and the immunoprecipitated fraction (30%) were then blotted for FLAG, Itch, WWP1, or WWP2. α-tubulin was used as a loading control. **h** COS-1 cells that express siRNA-resistant Myc-evectin-2 WT or PPPA mutant in a cumate-dependent fashion were established. These cells were first treated with control or evectin-2 siRNA for 24 h, and incubated with 30 μM cumate for another 24 h. The cells were then fixed, permeabilized, and stained for YAP and Myc. Arrows indicate cells that express Myc-evt-2 WT or PPPA. **i** Subcellular localization of YAP in cells in **h** was examined. **j** FLAG-WWP2 and Myc-evectin-2 were expressed in HEK293T cells for 24 h. Cell lysates with 1% Triton X-100 were immunoprecipitated with anti-FLAG antibody. The immunoprecipitates were then blotted for ubiquitin (Ub), FLAG, or Myc. α-tubulin was used as a loading control. **k** A model of the regulation of YAP on RE membranes in proliferating cells. Data are mean ± s.d. from two (for **d**, n > 40 cells; for **i**, n > 30 cells) or three (for **e**) independent experiments. Statistical significance was determined with one-way analysis of variance followed by Tukey–Kramer post hoc test (for **e**) or Kruskal–Wallis test followed by Steel–Dwass post hoc test (for **d**, **i**); **P < 0.01, ***P < 0.001, NS not significant. The nuclei were stained with DAPI. Scale bars, 10 μm

**Protein identification by mass spectrometry**. Biotinylated proteins were dissociated from Dynabeads with 50 mM Tris-HCl buffer (pH 7.5) containing 150 mM NaCl, 2% SDS, and 3 mM biotin at 98 °C for 5 min. After methanol/chloroform precipitation, the biotinylated protein pellet was resolubilized with 100 mM ammonium bicarbonate (AmBic, pH 8.8) containing 7 M guanidine hydrochloride and 0.01% decyl β-D-glucopyranoside (DG). The resolubilized proteins were reduced by 5 mM TCEP at 65 °C for 30 min, and then alkylated by 10 mM iodoacetamide at 25 °C for 30 min. Alkylated samples were diluted fivefold in 100 mM AmBic (pH 8.8) and 0.01% DG buffer containing 500 ng lysyl endopeptidase. After incubating at 37 °C for 3 h, the samples were diluted twofold in $H_2O$ containing 100 ng trypsin and digested at 37 °C for 16 h. The digested samples were acidified with trifluoro acetic acid and purified on C18 spin column. The purified peptides were dried in a vacuum concentrator and resolved with 0.1% formic acid (FA).

In brief, the resulting peptides were analyzed by nano-flow liquid chromatography coupled to tandem mass spectrometry (LC-MS/MS) with a custom-made nano-pump system (LC-Assist)/TripleTOF 5600 + spectrometer (Sciex)[43]. The samples were loaded on a custom-made C18 column (150 μm × 50 mm) packed with Mightysil RP-18 GP (3 μm) (Kanto Chemical). For elution, mobile phase A (0.1% FA in $H_2O$) and mobile phase B (0.1% FA in acetonitrile) were used with a 120 min linear gradient from 0 to 40% B at a flow rate of 100 nl $min^{-1}$. Eluted peptides from the reversed-phase chromatography were directly loaded on the nESI source in positive ionization and high sensitivity modes. The MS survey spectrum was acquired in the range of 400–1500 $m/z$ in 250 ms. For information-dependent acquisition, the 25 precursor ions above 50 counts threshold with charge states +2 and +3 were selected for MS/MS scans. Each MS/MS experiment set the precursor $m/z$ on a 12 s dynamic exclusion, and scan range was 100–1500 $m/z$ in 100 ms.

Protein identifications were performed with the ProteinPilot software (Sciex) using the NCBI non-redundant human protein data set (NCBI-nr RefSeq Release 71, containing 179460 entries). The tolerances were specified as ±0.05 Da for peptides and ±0.05 Da for MS/MS fragments. Identified peptides (confidence >95%) and proteins were listed in Supplementary Data 1. Relative quantification of identified proteins was performed based on iBAQ (intensity-based absolute quantification) method[44] without calibration with the Universal Proteomics Standard. The value of each protein (iBQ value) is listed in Supplementary Data 1.

**Immunocytochemistry**. Cells were washed with PBS briefly, fixed with 4% paraformaldehyde (PFA) in PBS at room temperature and permeabilized with 0.1% Triton X-100 in PBS for 5 min at room temperature. Blocking was performed with 3% BSA in PBS at room temperature for 30 min. The cells were then incubated with primary antibodies diluted in 3% BSA in PBS (for PFA-fixed cells) for 12–16 h. After washing three times with PBS, the cells were incubated with secondary antibodies at room temperature for 1 h, washed three times with PBS, and then mounted in PermaFluor (Thermo). Nuclei were stained with DAPI (D9542, Sigma). For biotinylation, the cells were incubated for 24 h in complete media supplemented with or without 50 μM biotin.

**Immunoprecipitation**. COS-1 cells grown on 100-mm dishes were transfected with FLAG-tagged evectin-2 (WT, PPPA, or ΔPPPY) or empty vector (pcDNA3-FLAG) using Lipofectamine 2000 (Invitrogen). At 24 h after transfection, the cells were lysed in 1 ml of immunoprecipitation buffer (25 mM HEPES-KOH pH 7.33, 150 mM NaCl, 1 mM EDTA, and 1% Triton X-100, cOmplete Protease inhibitor cocktail (Roche)). The cell lysates were centrifuged at 15,000×g for 20 min at 4 °C, and the resultant supernatants were incubated for 2 h at 4 °C with anti-FLAG M2 affinity gel beads (Sigma). The beads were washed three times with immunoprecipitation buffer. Immunoprecipitated proteins were analyzed by western blotting.

**qRT-PCR**. Total RNA was extracted from COS-1, PSA-3, or MDA-MB-231 cells using Isogen II (Nippon EGT) and reverse-transcribed using the High Capacity cDNA Reverse Transcription kit (Applied Biosystems). Quantitative real-time PCR (qRT-PCR) was performed using SYBR Green PCR Master Mix (TaKaRa) and a Light-Cycler 480 (Roche Diagnostics). The primers used in the present study are listed in Supplementary Table 2.

**Confocal microscopy**. Confocal microscopy was performed using a TCS SP8 (Leica) with a 63 × 1.2 Plan-Apochromat water immersion lens.

**Western blotting**. Proteins were separated in 10–12% polyacrylamide gel and then transferred onto polyvinylidene difluoride membranes (Millipore). The membranes were blocked for 30 min using 10 mM Tris-HCl pH 7.4, 150 mM NaCl, 0.05% Tween-20, and 5% low-fat milk powder. The membranes were incubated with primary antibodies, followed by secondary antibodies conjugated to HRP. The proteins were visualized by chemiluminescence using an ImageQuant LAS4000 analyzer (GE Healthcare). Uncropped images are shown in Supplementary Fig. 9.

**Luciferase assay**. COS-1 cells seeded on 24-well plates were transiently transfected with 8xGTIIC TEAD luciferase reporter plasmid (100 ng), pRL-TK (10 ng) as internal control, and GFP-2xPH WT or K20E-expression plasmid in pEGFP-C2 vector (100 ng). At 24 h after transfection, the luciferase activity in the total cell lysate was measured.

**Cell proliferation assay**. A total of $1.2 \times 10^5$ MDA-MB-231 cells were seeded into a six-well plate in triplicate. After 24 h, cells were treated with siRNA. After 48 h, $5 \times 10^4$ cells were replated into a 12-well plate in triplicate. After 24 h, cells were further treated with the same siRNA. The cells were counted every 24 h after replating.

**PSA-3 cells**. For PS depletion, the PSA-3 cells were cultured in a medium supplemented with 6% dFBS for 72 h. To restore the levels of PS, 40 μM ethanolamine was added to the culture.

**Fluorescent image analysis**. Quantification of images was performed with ImageJ software (NIH). The fluorescence intensity and Pearson coefficients were determined with the RGB Profile Plot and JACoP plugins, respectively.

**Statistical analysis**. Error bars displayed throughout this study represent s.d. Statistical analysis was performed using two-tailed Student's $t$ test, Steel–Dawass test, or one-way analysis of variance followed by Tukey–Kramer post hoc test.; *$P$ < 0.05; **$P$ < 0.01; ***$P$ < 0.001; NS not significant ($P$ > 0.05).

**Data availability**. The data that support the findings of this study are available from the corresponding author on request.

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

## Acknowledgements

We thank J. McKenzie (Lawrence Berkeley National Laboratory) for comments on the manuscript and C. Inoue (the University of Tokyo) for her assistance. This work was supported by JSPS KAKENHI Grant Numbers JP16H04782 (To.T.), JP15H05903 (To.T.), JP17H06164 (H.A.), JP17H06418 (H.A.), and JP17K15445 (K.M.); AMED-CREST (15652265) (H.A.); the Science and Technology Research Promotion Program for Agriculture, Forestry, Fisheries and Food Industry from the Ministry of Agriculture, Forestry and Fisheries of Japan (15650430) (H.A.); a Grant-in-Aid from the Uehara Memorial Foundation (H.N.); JSPS Research Fellowship for Young Scientists (DC1, 12380) (T.M.); and Ono Pharmaceutical Co., Ltd. (H.A.).

## Author contributions

T.M. designed and performed experiments, analyzed data, interpreted results, and wrote the paper; K.M. designed and performed experiments, analyzed data, and interpreted results; Ta.N. performed experiments with Nedd4 E3 ligases; J.H., T.H., S.I., and To.N. performed proteomics analysis; N.M. and H.N. interpreted results; J.N. and K.S. designed experiments with MDA-MB-231 cells; Ta.T and S.M. designed the BioID experiments; and To.T. and H.A. designed experiments, interpreted results, and wrote the paper.

## Additional information

**Competing interests:** The authors declare no competing financial interests.

