## [Peer Review File · Nature Communications]

Reviewers' Comments:

Reviewer #1 (Remarks to the Author)

In this study, Matsudaira et al reported the regulation of LATS and YAP by phosphatidylserine at the recycling endosomes (RE). Proximity biotinylation, mass spectrometry, immunofluorescence experiments showed that YAP is associated with RE. Knockdown of RE proteins or manipulation of phosphatidylserine levels affects YAP phosphorylation and subcellular localization. The authors also presented data indicating that the RE protein evectin-2 activates WWP1/2 E3 ligases to target LATS for degradation. The observation that phosphatidylserine may regulate the Hippo pathway is potentially interesting. However, several key questions need to be addressed before one can conclude that phosphatidylserine indeed plays a significant physiological role in the regulation of LATS and YAP.

Comments

The first part of the study of YAP association with RE and the second part of the study of phosphatidylserine in LATS protein level are disconnected. What is the relationship between YAP localization on RE and LATS regulation by evectin-2? At least some discussions are needed.

A few major questions need to be addressed. Is the effect of phosphatidylserine on YAP mediated by LATS? The authors should perform ATP8A1 knockdown and GFP-2xPH experiments (Fig.2) in LATS1 and LATS2 double knockout cells. Does phosphatidylserine regulate LATS stability? LATS ubiquitination, protein stability should be determined in cells with altered phosphatidylserine levels and with evt-2 knockdown. LATS activity is potently regulated by phosphorylation. Do manipulation of phosphatidylserine and RE proteins affect LATS phosphorylation/activity?

Is TAZ, the YAP homolog, associated with RE?

Fig.2a,b. Why YAP is not on RE at high density? Does density affect the phosphatidylserine levels on RE? The ATP8A1 knockdown efficiency should be shown.

Fig.2h. The data is not informative because the dialyzed serum may have lost many factors important for YAP activation but completely unrelated to phosphatidylserine. This experiment does not support a role of phosphatidylserine in YAP regulation. The authors need to perform additional experiments to alter phosphatidylserine levels and then determine the effect on YAP phosphorylation and localization. LATS protein levels should be determined in Fig.2d and 2h.

Fig.2f. Does the GFP-2XPH affect YAP phosphorylation and localization?

Fig.3h. The authors need to show that the growth inhibitory effect by evt-2 siRNA is mediated by YAP inhibition.

Fig.4a. The co-IP data is hard to believe. How could the co-IPed Itch, WWP1, and WWP2 be more enriched in the "ppt" by the FLAG-evt-2 IP when compared to their corresponding "input" (lysate)?

Fig.4c. The effects on LATS protein levels and YAP phosphorylation are very modest. WWP1 and WWP2 double knockout should be included.

Fig.4j. Experiment needs to be repeated with similar level expression of evt-2 WT and PPPA mutant.

Reviewer #2 (Remarks to the Author)

In this paper Matsudaira et al. describe a connection between phosphatidylserine (PS) in recycling endosomes (RE) and the transcriptional regulator YAP. The authors use a PS-binding construct fused to BirA* to biotinylate proteins that are in close proximity to PS and identify several proteins that are known regulators of YAP signaling including the YAP protein itself. They find that nuclear localization of YAP requires the PS binding protein, evelctin2 and the PS flippase, ATP8A1, both being enriched in the recycling compartment. The authors also show that evelctin-2 associates with members of the Nedd4 ubiquitin E3 ligases, which control the degradation of a Lats1, a kinase that phosphorylates YAP thereby keeping it out of the nucleus and directing it to degradation. They identify a proline-rich sequence in evelctin-2 that is critical for interaction with the Nedd4 E3 ligases and show that unlike the wild-type evelctin-2, the proline-mutant is unable to rescue the YAP phenotype in evelctin-2 depleted cells.

Finally the authors show that the growth of a cancer cell line whose proliferation is YAP/TAZ dependent can be inhibited by knocking down evelctin-2 or the ATP8A1 flippase.

These are interesting findings that describe a new connection between PS and the transcriptional regulator YAP. YAP/TAZ signaling is a hot topic with many open questions and this study adds an important new regulatory paradigm. The experiments are straight-forward and well-documented, and for the most part support the conclusions. My major overall criticism is that we do not learn whether the PS-regulation is simply permissive or it is part of the control how cell density is translated to YAP/TAZ signaling. Also, it is not clear to this Reviewer why the 2x-PH evelctin-2 shows the restricted localization in the RE. This constructs is supposed to find PS in every compartment, most prominently in the plasma membrane.

There are several specific questions that should be addressed before a decision is made regarding suitability for publication in Nature Communications.

1. The 2x-PH of EVELCTIN-2 has been used as a PS reporter showing prominent plasma membrane localization (Chung et al, PMID: 26206935). It is very curious why 2xPH-evelctin-2 does not show the normal PS distribution in the present study? (PM, RE, LE?).
2. It is not clear based on the authors's findings what the sequence of events is regarding YAP activation or degradation. Is YAP phosphorylated at the RE by Lats1? Is Lats1 found in the RE (no data on this in the paper)? Once phosphorylated where does YAP go to be degraded as suggested by the cartoon.
3. It is not clear how -phospho YAP is showing up if it is destined for degradation. Shouldn't the level of YAP decrease under conditions when its ~P is increased?
4. Where are the data to show that evelctin-2 brings the WWP1/2 and Itch to the RE? The co-immuno-precipitation suggests an interaction but it is not shown that w/o evelctin these proteins do not localize to the RE.
5. What is the localization of Lats1 in the cell and how does that change when PS is not present on the RE?
6. YAP/TAZ signaling depends greatly on cell confluence. How does the PS regulation relate to the cell density-driven regulation?

To the reviewers: We are very grateful for your comments. They have helped us to greatly improve the ms.

Reviewer1:

In this study, Matsudaira et al reported the regulation of LATS and YAP by phosphatidylserine at the recycling endosomes (RE). Proximity biotinylation, mass spectrometry, immunofluorescence experiments showed that YAP is associated with RE. Knockdown of RE proteins or manipulation of phosphatidylserine levels affects YAP phosphorylation and subcellular localization. The authors also presented data indicating that the RE protein evectin-2 activates WWP1/2 E3 ligases to target LATS for degradation. The observation that phosphatidylserine may regulate the Hippo pathway is potentially interesting. However, several key questions need to be addressed before one can conclude that phosphatidylserine indeed plays a significant physiological role in the regulation of LATS and YAP.

Comments

The first part of the study of YAP association with RE and the second part of the study of phosphatidylserine in LATS protein level are disconnected. What is the relationship between YAP localization on RE and LATS regulation by evectin-2? At least some discussions are needed.

> As shown in Fig. 2a in the original manuscript, anti-YAP antibody that was raised against 12 amino acids at the C-terminus of YAP [Y. Bao *et al.*, *J. Biochem.* 150, 199-208 (2011)] stained REs. During the revision process, we found that anti-phosphorylated YAP (p-YAP) antibody evenly stained the cytoplasm in low-density cells (see below), indicating that unphosphorylated YAP was preferentially associated with REs.

Our original data showed that evectin-2/Nedd4 E3 ligases, RE-associated proteins, reduced the protein level of Lats1 (Fig. 4c and Supplementary Fig. 5 in the original manuscript). We also found that the phosphorylated Lats1 was increased by knockdown of ATP8A1, a PS-flippase at REs (Fig. 2f in the revised manuscript), while total Lats1 was not altered. Thus, REs function in the inactivation of Lats1 by both dephosphorylation and degradation, and YAP at REs may hardly be phosphorylated by Lats1.

At present, the mechanism of the increased phosphorylated Lats1 by knockdown

of ATP8A1 is not known.

The following texts were included in the revised manuscript.

page 5, line 9↑:

"YAP can be phosphorylated on Ser127 by Lats kinases (Lats1 and Lats2)². Phosphorylated YAP binds to 14-3-3 proteins in the cytoplasm, which prevents the translocation of YAP to the nucleus^{2,3}. Although YAP was found to be enriched at REs in low-density cells (Fig. 2a), phosphorylated YAP was evenly distributed throughout the cytoplasm (Fig. 2b). The result suggested that YAP at REs was preferentially in an unphosphorylated form (Fig. 2b). As expected, phosphorylated YAP was excluded from the nucleus."

page 9, line 4↑:

"Our proteomics analysis of proteins proximal to PS revealed a new role of the RE membranes in YAP activation and cell proliferation: in low-density cells, ATP8A1 and evectin-2/Nedd4 E3 ligases contribute to reduce the level of phosphorylated Lats1 and Lats1, respectively. The limited Lats1 function at REs in low-density cells is consistent with the observation that YAP present at REs is preferentially in an unphosphorylated form (Fig. 2b)."

A few major questions need to be addressed. Is the effect of phosphatidylserine on YAP mediated by LATS? The authors should perform ATP8A1 knockdown and GFP-2xPH

experiments (Fig.2) in *LATS1* and *LATS2* double knockout cells. Does phosphatidylserine regulate *LATS* stability? *LATS* ubiquitination, protein stability should be determined in cells with altered phosphatidylserine levels and with *evt-2* knockdown. *LATS* activity is potently regulated by phosphorylation. Do manipulation of phosphatidylserine and RE proteins affect *LATS* phosphorylation/activity?

> (a) To address whether the effect of phosphatidylserine on YAP is mediated by Lats, we compared the level of p-YAP between ATP8A1-knockdown cells and ATP8A1/Lats1/Lats2-triple knockdown cells. As shown in the revised Supplementary Fig. 3, p-YAP level was reduced by Lats1/2 knockdown, suggesting that the increase of pYAP caused by ATP8A1 knockdown was indeed mediated by Lats proteins. We could not validate the effects of GFP-2xPH overexpression on p-YAP in Lats1/2-knockdown cells, because most of the cells were detached from the culture dish after double transfections (siRNA and plasmids were transfected at the different times).

(b) The stability of Lats1 protein was examined in ATP8A1-knockdown cells and *eventin-2*-knockdown cells. We treated the cells with cycloheximide to inhibit protein translation, and monitored the level of Lats1 protein by WB. As shown in the images below, the stability of Lats1 was not different between ATP8A1-knockdown cells and control cells. In contrast, the stability of Lats1 was enhanced in *eventin-2*-knockdown cells.

We then examined the ubiquitination of Lats1. As shown in the images below, we detected ubiquitinated Lats1 in cells that were treated with the proteasome inhibitor MG132 for 4 and 8 hours. Importantly, the ubiquitination of Lats1 was mostly abolished by *eventin-2* knockdown, but not by ATP8A1 knockdown. The involvement of *eventin-2* in Lats1 ubiquitination is consistent with the above observation that the

stability of Lats1 is enhanced by evectin-2 knockdown.

(c) As indicated to the response to the previous question, ATP8A1 knockdown or cellular PS depletion increased the level of phosphorylated Lats1, but did not affect the level of Lats1 (the revised Fig. 2f and 3f). In contrast, evectin-2 knockdown increased the level of Lats1 and phosphorylated Lats1 (the revised Fig. 4g).

From these results, we are now assuming that evectin-2 mediates the ubiquitination and degradation of Lats1, whereas the concentrated PS in REs contributes to the dephosphorylation but not to the degradation of Lats1. We previously showed that PS was still present in the cytosolic leaflet of RE membranes in ATP8A1-depleted cells and evectin-2 was still localized at REs [S. Lee et al., *EMBO J.* 34, 669-688 (2015)]. Thus, ATP8A1 depletion and evectin-2 depletion may not necessarily result in the same outcome.

With these new results, the following texts were included in the revised manuscript.

page 5, line 2↑:

"ATP8A1 is a PS-flippase that localizes to REs. As a flippase, it catalyzes the enrichment of PS in the cytoplasmic leaflet of RE membranes⁸. We found that knockdown of ATP8A1 with siRNA 1) significantly reduced the nuclear localization of YAP (Fig. 2c-e), 2) increased the phosphorylation of YAP (Fig. 2f), 3) increased the phosphorylation of Lats1 on Ser909, an activated form of Lats1²³ (Fig. 2f), and 4)

significantly decreased the mRNA expression of CTGF, a YAP-regulated gene (Fig. 2g). The increase of phosphorylated YAP was reduced by depletion of Lats1/2 (Supplementary Fig. 3). These results indicated that the concentrated PS in the cytoplasmic leaflet of RE membranes is critical for the nuclear localization of YAP and the transcription of YAP-downstream genes by suppressing Lats1 activity."

page 6, line 10↑:

"Second, YAP signalling was examined in PS-auxotrophic mutant CHO cells (PSA-3)²⁵ that lack the activity of phosphatidylserine synthase-1. When PSA-3 cells are cultured with dialyzed fetal bovine serum (dFBS) that lacks ethanolamine (Etn), cellular levels of PS decrease by around 30% compared to PSA-3 cells cultured with dFBS plus Etn²⁶. The PS-deficient culture conditions significantly reduced mRNA expression of CTGF (Fig. 3e) and increased the level of phosphorylated YAP and phosphorylated Lats1 (Fig. 3f). In contrast, including Etn in the culture medium to maintain cellular levels of PS²⁶ did not affect the mRNA expression of CTGF, or the level of phosphorylated YAP or phosphorylated Lats1 (Fig. 3e,f)."

page 7, line 7↑:

"Knockdown of evecitin-2 in MDA-MB-231 cells significantly reduced mRNA expression of CTGF (Fig. 4f), increased the level of Lats1 and phosphorylated YAP (Fig. 4g), and reduced the cell proliferation rate (Fig. 4h and Supplementary Fig. 4d). Further knockdown of Lats proteins significantly restored the cell proliferation rate (Fig. 4i), suggesting that inhibition of proliferation by evecitin-2 knockdown was mediated by YAP inhibition."

page 8, line 11↑:

"We examined whether evecitin-2 contributes to the stability of Lats1. Cells were treated with cycloheximide (CHX), an agent that inhibits protein translation, and the level of Lats1 was monitored. In control cells, the amount of Lats1 was decreased by 40% after 12 h treatment with CHX (Supplementary Fig. 6a). In contrast, in evecitin-2-knockdown cells, the amount of Lats1 was decreased by only 15%. These results indicated that stability of Lats1 was negatively regulated by evecitin-2. We then examined the ubiquitination of Lats1. In control cells, after 4 and 8 h treatment with the proteasome

inhibitor MG132, ubiquitination of Lats1 was detected (Supplementary Fig. 6b). Importantly, the ubiquitination of Lats1 was mostly abolished by evecin-2 knockdown. These results suggested that evecin-2 negatively regulates the stability of Lats1 through the ubiquitination of Lats1. Knockdown of ATP8A1, which did not affect the amount of Lats1 (Fig. 2f), did not affect the stability of Lats1 (Supplementary Fig. 6a) or the ubiquitination of Lats1 (Supplementary Fig. 6b)."

Is TAZ, the YAP homolog, associated with RE?

> We did not find TAZ association with REs. Flag-Myc-TAZ (WT or 4SA constitutive active mutant) was expressed in low-density COS-1 cells and its localization was examined (see the images below). We found that TAZ (WT) was observed mainly in the cytoplasm, whereas TAZ (4SA) was enriched in the nucleus, but not at REs in these cells.

YAP contains a proline-rich NH₂-terminus that is absent in TAZ. YAP has two WW-domains, whereas TAZ has only one. A recent study indentified a number of proteins (P. Kohki et al., *Am. J. Physiol. Cell Physiol.* 306, C805 (2014)) that specifically bind YAP or TAZ, respectively. We assume that the difference of interacting proteins between YAP and TAZ may account for the different subcellular localizations of YAP and TAZ.

Fig.2a,b. Why YAP is not on RE at high density? Does density affect the phosphatidylserine levels on RE? The ATP8A1 knockdown efficiency should be shown.

> As shown in the images below, we found that PS mostly localized at REs even in high-density cells. Therefore, we cannot attribute the dissociation of YAP from REs in high-density cells simply to the change of PS levels at REs.

We previously showed that PS at REs is essential for membrane traffic through REs [S. Lee et al., *EMBO J.* 34, 669-688 (2015)]. Because even cells in high density continuously endocytose and recycle molecules such as transferrin to obtain iron from the culture medium, we assume that the cells still need to enrich PS to REs to proceed the endosomal membrane traffic.

We showed that unphosphorylated YAP is associated with REs and this association was abolished by ATP8A1 knockdown, indicating that the concentrated PS on the surface of REs is critical for YAP association with REs. However, at present, we do not know how YAP is recruited to REs in low-density cells, because known PS-binding motifs are not present in YAP. The recruitment of YAP to REs could be mediated through proteins that interact with YAP and localize at REs, such as Amot family proteins [(B. Zhao et al., *Genes Dev.* 25, 51-63 (2011)) [B. Heller et al., *J. Biol. Chem.* 285, 12308-12320 (2010)]. Such interaction might be impaired in high-density cells.

Knockdown efficiency of ATP8A1 in the original Fig. 2b was provided in the original Supplementary Fig. 3a. In the revised manuscript, we moved the knockdown efficiency to Fig. 2c.

Fig.2h. The data is not informative because the dialyzed serum may have lost many factors important for YAP activation but completely unrelated to phosphatidylserine. This experiment does not support a role of phosphatidylserine in YAP regulation. The authors need to perform additional experiments to alter phosphatidylserine levels and then determine the effect on YAP phosphorylation and localization. LATS protein levels should be determined in Fig.2d and 2h.

>We performed an additional experiment to alter the PS levels in the cytoplasmic leaflet. We made use of overexpression of phosphatidylserine decarboxylase 1 (PSD1), an enzyme that degrades PS, based on a recent paper [O. Onguka *et al.*, *J. Biol. Chem.* 290, 12744-12752 (2015)]. As shown in the revised Fig. 3g, overexpression of PSD1 abolished the RE localization of the PS probe 2xPH, while overexpression of catalytically inactive PSD1 (S463A) did not affect the RE localization of 2xPH, suggesting that overexpression of PSD1 reduced PS in REs.

We then examined the effect of PSD1 overexpression on YAP. As shown in the revised Fig. 3h-j, the overexpression of PSD1, but not PSD1 (S463A), suppressed the nuclear translocation of YAP and increased the level of phosphorylated YAP. These results provide further evidence that PS has a role in YAP activation.

As suggested, Lats protein levels were determined in the experiment of ATP8A1 knockdown and PS-deficient PSA-3 cells. We also examined phosphorylated Lats1, which is related to the question 2. As shown in the revised Fig. 2f and the revised Fig. 3f, we found that knockdown of ATP8A1 or PS depletion in PSA-3 cells did not affect the level of Lats proteins, but instead the level of phosphorylated Lats1, the activated form of Lats1. These results suggested that phosphatidylserine levels contribute positively to YAP signalling pathway by reducing the level of activated Lats1. The level of Lats proteins was not determined in the PSD1 experiments because only 20-30% of the cells were transfected with PSD1 under the conditions used.

The following texts were included in the revised manuscript.

page 5, line 2↑:

"ATP8A1 is a PS-flippase that localizes to REs. As a flippase, it catalyzes the enrichment of PS in the cytoplasmic leaflet of RE membranes⁸. We found that

knockdown of ATP8A1 with siRNA 1) significantly reduced the nuclear localization of YAP (Fig. 2c-e), 2) increased the phosphorylation of YAP (Fig. 2f), 3) increased the phosphorylation of Lats1 on Ser909, an activated form of Lats1²³ (Fig. 2f), and 4) significantly decreased the mRNA expression of CTGF, a YAP-regulated gene (Fig. 2g). The increase of phosphorylated YAP was reduced by depletion of Lats1/2 (Supplementary Fig. 3). These results indicated that the concentrated PS in the cytoplasmic leaflet of RE membranes is critical for the nuclear localization of YAP and the transcription of YAP-downstream genes by suppressing Lats1 activity."

page 6, line 10↑:

"Second, YAP signalling was examined in PS-auxotrophic mutant CHO cells (PSA-3)²⁵ that lack the activity of phosphatidylserine synthase-1. When PSA-3 cells are cultured with dialyzed fetal bovine serum (dFBS) that lacks ethanolamine (Etn), cellular levels of PS decrease by around 30% compared to PSA-3 cells cultured with dFBS plus Etn²⁶. The PS-deficient culture conditions significantly reduced mRNA expression of CTGF (Fig. 3e) and increased the level of phosphorylated YAP and phosphorylated Lats1 (Fig. 3f). In contrast, including Etn in the culture medium to maintain cellular levels of PS²⁶ did not affect the mRNA expression of CTGF, or the level of phosphorylated YAP or phosphorylated Lats1 (Fig. 3e,f)."

page 6, line 2↑:

"Third, we made use of overexpression of phosphatidylserine decarboxylase 1 (PSD1), a mitochondrial enzyme that degrades PS²⁷⁻²⁹. We generated yeast PSD1 mutant (yPSD1ΔN) that lacks a *N*-terminal mitochondria-targeting sequence and a transmembrane domain. The RE localization of the PS probe 2xPH was drastically lost in cells that overexpress yPSD1ΔN, but not catalytically inactive yPSD1ΔN (S463A) (Fig. 3g), indicating that overexpression of yPSD1ΔN reduced PS in REs. We then examined the effect of yPSD1ΔN overexpression on YAP. The expression of yPSD1ΔN, but not yPSD1ΔN (S463A), suppressed the nuclear translocation of YAP (Fig. 3h,i) and increased the level of phosphorylated YAP (Fig. 3j). These results provide further evidence that PS has a role in YAP activation."

Fig.2f. Does the GFP-2XPH affect YAP phosphorylation and localization?

> As shown in the revised Fig. 3a-c, overexpression of GFP-2xPH suppressed the nuclear translocation of YAP and increased the level of phosphorylated YAP. In contrast, overexpression of GFP-2xPH (K20E), which does not bind PS [S. Lee et al., *EMBO J.* 34, 669-688 (2015)], had no effect. Therefore, like the depletion of PS, masking of PS in the cytosolic leaflet caused the inactivation of YAP.

The following texts were included in the revised manuscript.

page 6, line 9:

"First, the PS-specific probe GFP-2xPH was used to mask the PS at the cytosolic leaflet of RE membranes⁸. Overexpression of GFP-2xPH in the cytoplasm significantly reduced the nuclear localization of YAP (Fig. 3a,b), increased the level of phosphorylated YAP (Fig. 3c), and reduced the transcriptional activity of YAP (Fig. 3d) using the TEAD reporter system²⁴. In contrast, overexpression of the mutant (GFP-2xPH K20E)⁸, which does not bind PS, had no effect (Fig. 3a-d)."

Fig.3h. The authors need to show that the growth inhibitory effect by evt-2 siRNA is mediated by YAP inhibition.

> We examined whether knockdown of Lats restores the growth inhibition caused by knockdown of evectin-2. As shown in the revised Fig. 4i, knockdown of Lats1 or Lats1/2 significantly restored the growth of evectin-2-depleted cells. These results suggested that the growth inhibition by evectin-2 knockdown was mediated by YAP inhibition.

The following texts were included in the revised manuscript.

page 7, line 7↑:

"Knockdown of evectin-2 in MDA-MB-231 cells significantly reduced mRNA expression of CTGF (Fig. 4f), increased the level of Lats1 and phosphorylated YAP (Fig. 4g), and reduced the cell proliferation rate (Fig. 4h and Supplementary Fig. 4d). Further knockdown of Lats proteins significantly restored the cell proliferation rate (Fig. 4i), suggesting that inhibition of proliferation by evectin-2 knockdown was mediated by YAP inhibition."

Fig.4a. The co-IP data is hard to believe. How could the co-IPed Itch, WWP1, and WWP2 be more enriched in the “ppt” by the FLAG-evt-2 IP when compared to their corresponding “input” (lysate)?

> We used 0.8% of the lysate as an input and 10% of the ppt fraction in this experiment, therefore the ppt fraction was enriched by 12.5-fold more than the input fraction. We included the information in the revised Fig. 5a and the corresponding Figure legend. Similarly, we revised the legend to the revised Fig. 5g.

Fig.4c. The effects on LATS protein levels and YAP phosphorylation are very modest. WWP1 and WWP2 double knockout should be included.

> We carried out double knockdown of WWP1 and WWP2. As shown in the revised Fig. 5e and 5f, we did not see an augmented effect of the double knockdown on levels of p-YAP, Lats1, or CTGF. Given that WWP1 and WWP2 form a complex to ubiquitinate specific proteins for degradation (Chaudhary N *et al.*, Mol. Cell Biol. 2014, **34**, 3754-3764), it is possible that the WWP1/WWP2 complex is also involved in the Lats1 degradation. In this scenario, the knockdown of either WWP1 or WWP2 would have the same effect as double knockdown of WWP1 and WWP2.

The increase of Lats1 by WWP1 and/or WWP2 knockdown appeared to correspond to that of Lats1 by evectin-2 knockdown (the revised Fig. 4g). This supports our proposed model (the revised Fig 5k), in which evectin-2 mediates the degradation of Lats1 through Nedd4 E3 ligases.

The following texts were included in the revised manuscript.

page 8, line 8:

"Knockdown of WWP1 or WWP2 in COS-1 cells decreased the nuclear localization of YAP (Fig. 5c,d) and mRNA expression of CTGF (Fig. 5e), and increased the levels of Lats1 and phosphorylated YAP (Fig. 5f). Double knockdown of WWP1 and WWP2 did not have an additive effect on the levels of Lats1, phosphorylated YAP, or CTGF."

Fig.4j. Experiment needs to be repeated with similar level expression of evt-2 WT and PPPA mutant.

> Please see the input in original Fig. 4j. The expression levels of Myc-tagged evectorin-2 (WT) and Myc-tagged evectorin-2 (PPPA) were mostly similar. However, because the PPPA mutant did not bind to WWP2 (original Fig. 4g), the amount of Myc-tagged evectorin-2 (PPPA) immunoprecipitated with FLAG-WWP2 was much less than that of Myc-tagged evectorin-2 (WT).

Reviewer2:

In this paper Matsudaira et al. describe a connection between phosphatidylserine (PS) in recycling endosomes (RE) and the transcriptional regulator YAP. The authors use a PS-binding construct fused to BirA to biotinylate proteins that are in close proximity to PS and identify several proteins that are known regulators of YAP signaling including the YAP protein itself. They find that nuclear localization of YAP requires the PS binding protein, evectin2 and the PS flippase, ATP8A1, both being enriched in the recycling compartment. The authors also show that evectin-2 associates with members of the Nedd4 ubiquitin E3 ligases, which control the degradation of a Lats1, a kinase that phosphorylates YAP thereby keeping it out of the nucleus and directing it to degradation. They identify a proline-rich sequence in evectin-2 that is critical for interaction with the Nedd4 E3 ligases and show that unlike the wild-type evectin-2, the proline-mutant is unable to rescue the YAP phenotype in evectin-2 depleted cells. Finally the authors show that the growth of a cancer cell line whose proliferation is YAP/TAZ dependent can be inhibited by knocking down evectin-2 or the ATP8A1 flippase.*

These are interesting findings that describe a new connection between PS and the transcriptional regulator YAP. YAP/TAZ signaling is a hot topic with many open questions and this study adds an important new regulatory paradigm. The experiments are straight-forward and well-documented, and for the most part support the conclusions. My major overall criticism is that we do not learn whether the PS-regulation is simply permissive or it is part of the control how cell density is translated to YAP/TAZ signaling. Also, it is not clear to this Reviewer why the 2x-PH evectin-2 shows the restricted localization in the RE. This constructs is supposed to find PS in every compartment, most prominently in the plasma membrane.

There are several specific questions that should be addressed before a decision is made regarding suitability for publication in Nature Communications.

1. The 2x-PH of Evectin-2 has been used as a PS reporter showing prominent plasma membrane localization (Chung et al, PMID: 26206935). It is very curious why

2xPH-evectin-2 does not show the normal PS distribution in the present study? (PM, RE, LE?).

> As shown in the image below, GFP-2xPH overexpressed in COS-1 cells clearly labeled the PM as well as the REs. The image appears similar to the one in Chung's paper (for example, the left panel in Fig. 4C). Note that the intensity of GFP-2xPH at REs was stronger than that of GFP-2xPH at the PM in COS-1 cells. This may simply reflect the fact that REs in COS-1 cells are heavily clustered in the perinuclear region, leading to higher membrane content per volume at the REs than at the PM.

[GFP-2xPH overexpressed in COS-1 cells: Arrows indicate the PM localization of GFP-2xPH]

In the present study, we established cell lines that stably express 2xPH at very low levels, since the overexpression of 2xPH was cytotoxic and suppressed YAP activation (Fig. 2f in the original manuscript). Under these conditions, the PM localization of 2xPH was almost invisible.

2. It is not clear based on the authors's findings what the sequence of events is regarding YAP activation or degradation. Is YAP phosphorylated at the RE by Lats1? Is Lats1 found in the RE (no data on this in the paper)? Once phosphorylated where does YAP go to be degraded as suggested by the cartoon.

> Although we responded to similar questions raised by the reviewer1, we respond to the reviewer2 here. As shown in Fig. 2a in the original manuscript, anti-YAP antibody that was raised against 12 amino acids at the C-terminus of YAP [Y. Bao *et al.*, *J. Biochem.* 150, 199-208 (2011)] stained REs. During the revision process, we found that anti-phosphorylated YAP (p-YAP) antibody evenly stained the cytoplasm in low-density cells (see below), indicating that unphosphorylated YAP (activated form of

YAP) was preferentially associated with REs.

Our original data showed that evectin-2/Nedd4 E3 ligases, RE-associated proteins, reduced the protein level of Lats1 (Fig. 4c and Supplementary Fig. 5 in the original manuscript). We also found that the phosphorylated Lats1 was increased by knockdown of ATP8A1, a PS-flippase at REs (Fig. 2f in the revised manuscript), while total Lats1 was not altered. Thus, REs function in the inactivation of Lats1 by both dephosphorylation and degradation, and YAP at REs may hardly be phosphorylated by Lats1.

We did not observe Lats1 in the REs (please see our responses to the question 5 of reviewer2). As mentioned above, Lats1 may be quickly degraded by RE-localized evectin-2/Nedd4 E3 ligases.

p-YAP was increased by knockdown of ATP8A1 or evectin-2/NEDD4-E3 ligases. Regarding the fate of pYAP, it is known that pYAP binds to 14-3-3, which results in the sequestration of pYAP in the cytoplasm (J. Dong *et al.*, *Cell*, PMID 17889654; B. Zhao *et al.*, *Genes Dev.*, PMID 17974916) or the degradation by proteasome system (B. Zhao *et al.*, *Genes Dev.*, PMID 20048001; F. X. Yu *et al.*, *Cell*, PMID 26544935). It is possible that most of pYAP could be bound by cytosolic 14-3-3 and is not destined for the degradation under the present conditions.

We revised the following text to describe the cytosolic sequestration of p-YAP by

14-3-3. Since the essence of the Hippo pathway is the sequestration of p-YAP in the cytoplasm and not the degradation of p-YAP, we also revised the cartoon in Fig. 5 in the revised manuscript.

page 5, line 9↑:

"YAP can be phosphorylated on Ser127 by Lats kinases (Lats1 and Lats2)². Phosphorylated YAP binds to 14-3-3 proteins in the cytoplasm, which prevents the translocation of YAP to the nucleus^{2,3}."

3. It is not clear how –phospho YAP is showing up if it is destined for degradation. Shouldn't the level of YAP decrease under conditions when its ~P is increased?

> Please see the response to the question 2 of reviewer2.

4. Where are the data to show that evectorin-2 brings the WWP1/2 and Itch to the RE? The co-immuno-precipitation suggests an interaction but it is not shown that w/o evectorin these proteins do not localize to the RE.

> We examined whether evectorin-2 brings the Nedd4 E3 ligases to the REs. As shown in the image at the next page, evectorin-2 knockdown did not alter the RE localization of Itch, WWP1, or WWP2.

Then, we asked which part of the Nedd4 E3 ligases is required for the RE localization. We found that the Nedd4 E3 ligases lacking the C2 domain completely lost the RE localization and was distributed throughout the cytoplasm. Because the C2 domain is the domain that binds PS, the C2/PS interaction may be responsible for the RE localization of the Nedd4 E3 ligases.

As shown in the revised Fig. 5j, evectorin-2 rather functions in the activation of the Nedd4 E3 ligases through the interaction of the PPPY sequence of evectorin-2 and the WW domains of Nedd4 E3 ligases.

In the revised manuscript, we placed these results in Supplementary Fig. 8 and included the following text (page 9, line 10↑):

"Knockdown of evectin-2 did not alter the RE localization of Itch, WWP1, and WWP2 (Supplementary Fig. 8a). Therefore, evectin-2 functions in the activation of the Nedd4 E3 ligases at REs, but not in their recruitment to REs. The Nedd4 E3 ligases that lack N-terminal C2 domain did not localize to REs (Supplementary Fig. 8b), suggesting the critical role of the C2 domain in the recruitment of the Nedd4 E3 ligases to REs."

5. *What is the localization of Lats1 in the cell and how does that change when PS is not present on the RE?*

> This question is related to the question 2 of reviewer2. We examined the localization of Lats1 in COS-1 cells. As shown in the images below, GFP-Lats1 was distributed throughout the cytoplasm, with some accumulation at centrosome. The centrosomal localization of Lats1 was reported previously (*Sci Rep*, PMID 26530630). This subcellular localization did not change by ATP8A1 KD. As we stated in our response to

question 2 of reviewer2, Lats1 may be quickly degraded by RE-localized evectin-2/Nedd4 E3 ligases. Since Lats1 was not found on the REs, we would like not to include this issue in the revised manuscript.

6. *YAP/TAZ signaling depends greatly on cell confluence. How does the PS regulation relate to the cell density-driven regulation?*

> As shown in the images at the next page, PS mostly localized at REs even in high-density cells. Therefore, we cannot attribute the dissociation of YAP from REs in high-density cells simply to the change of PS levels at REs.

We previously showed that PS at REs is essential for membrane traffic through REs [S. Lee et al., *EMBO J.* 34, 669-688 (2015)]. Because even high-density cells endocytose and recycle molecules such as transferrin to obtain iron from culture medium continuously, we assume that the cells still need to enrich PS to REs to proceed the endosomal membrane traffic.

At present, we do not know how YAP is recruited to REs in low-density cells, because known PS-binding motifs are not present in YAP. The recruitment of YAP to REs could be mediated through proteins that interact with YAP and localize at REs, such as Amot family proteins (B. Zhao et al., *Genes Dev.* 25, 51-63 (2011)) (B. Heller et al., *J. Biol. Chem.* 285, 12308-12320 (2010)). Such interaction might be impaired in

high-density cells.

Since we found no altered PS distribution, the original title "Endosomal phosphatdylserine regulates the YAP sinalling pathway" may be misleading. We revised the title to "Endosomal phosphatdylserine is critical for the YAP activation in proliferating cells".

Reviewers' Comments:

Reviewer #1:

Remarks to the Author:

No further comments.

Reviewer #2:

Remarks to the Author:

This is the revised version of the Matsudaira paper on phosphatidylserine (PS) and YAP signaling.

The authors have performed additional experiments in response to the reviewers' comments. These experiments have provided additional insights and have slightly changed the conclusions and the mechanistic model compared to what was presented in the original version. Specifically, the authors now concluded that the Nedd4 E3 ligases (WWP1/2 and Itch) are not recruited by evectin-2 but via their C2 domains, presumably by binding PS. By inference the authors conclude that evectin-2 activates these E3 ligases rather than recruits them. However, although these are reasonable conclusions, they were not tested experimentally. It is also important to note that knock-down of the PS flippase ATP8A1 did not reproduce the effects of evectin-2 knock-down on Lats1 amount, phosphorylation or ubiquitination, whereas it reproduced the effects of evectin-2 knock down on YAP nuclear localization and signaling. Also significant is the finding that Lats1/2 distribution clearly does not indicate any special connection to PS, so if there is a connection that should affect only a spatially restricted pool of this enzyme. These results suggest that mechanistic connection between PS and YAP signaling is more complex than what is proposed by the authors in their mechanistic model.

The strength of the paper remains the clear documentation of a role of PS in YAP signaling that by itself is interesting. The authors have come a long way to experimentally determine the localization and functions of several potential regulators as suggested by their proximity biotinylation results. The authors have done several additional experiments in response to our requests and modified their conclusions based on the new results (including the title). The revised manuscript is stronger. My remaining problem is the mechanistic connection that the authors propose. I think the Discussion should reflect this ambiguity more clearly. It would add to the value of the paper rather than take away from it.

Reviewer2:

This is the revised version of the Matsudaira paper on phosphatidylserine (PS) and YAP signaling.

The authors have performed additional experiments in response to the reviewers' comments. These experiments have provided additional insights and have slightly changed the conclusions and the mechanistic model compared to what was presented in the original version. Specifically, the authors now concluded that the Nedd4 E3 ligases (WWP1/2 and Itch) are not recruited by evectin-2 but via their C2 domains, presumably by binding PS. By inference the authors conclude that evectin-2 activates these E3 ligases rather than recruits them. However, although these are reasonable conclusions, they were not tested experimentally. It is also important to note that knock-down of the PS flippase ATP8A1 did not reproduce the effects of evectin-2 knock-down on Lats1 amount, phosphorylation or ubiquitination, whereas it reproduced the effects of evectin-2 knock down on YAP nuclear localization and signaling. Also significant is the finding that Lats1/2 distribution clearly does not indicate any special connection to PS, so if there is a connection that should affect only a spatially restricted pool of this enzyme. These results suggest that mechanistic connection between PS and YAP signaling is more complex than what is proposed by the authors in their mechanistic model.

The strength of the paper remains the clear documentation of a role of PS in YAP signaling that by itself is interesting. The authors have come a long way to experimentally determine the localization and functions of several potential regulators as suggested by their proximity biotinylation results. The authors have done several additional experiments in response to our requests and modified their conclusions based on the new results (including the title). The revised manuscript is stronger. My remaining problem is the mechanistic connection that the authors propose. I think the Discussion should reflect this ambiguity more clearly. It would add to the value of the paper rather than take away from it.

>We agree that the mechanistic connection between PS and YAP signaling is more complex than what we originally proposed and that some of the conclusions were stated too strongly. To improve our manuscript we revised the manuscript as follows.

(i) In Supplementary Fig 8, we showed that evectin-2 knockdown did not affect the localization of three Nedd4 E3 ligases by microscopic analysis. However, we agree that this set of experiment does not rule out the possibility that evectin-2 recruits the Nedd4 E3 ligases. We toned down our statement that "evectin-2 does not recruit the ligases" to "evectin-2 does not appear to have a large role in their recruitment". Thus, we revised the sentences (page 9 line10↑ - page 9 line 5↑) in the former manuscript as follows.

“Knockdown of evectin-2 did not alter the RE localizations of Itch, WWP1, and WWP2 (Supplementary Fig. 8a), **implying that evectin-2 does not appear to have a large role in the recruitment of the Nedd4 E3 ligases to REs. Therefore, evectin-2 functions in the activation of the Nedd4 E3 ligases at REs, but not in their recruitment to REs.** The observation that the Nedd4 E3 ligases that lack the *N*-terminal C2 domain did not localize to REs (Supplementary Fig. 8b) suggested that the C2 domain has a role in the recruitment of the Nedd4 E3 ligases to REs. **Further studies will be needed to reveal the precise mechanism by which the Nedd4 E3 ligases are recruited to REs.**”

We also deleted the right part of Fig. 5k, because how the Nedd4 E3 ligases are recruited and activated in REs is not fully elucidated in the present study.

(ii) To reflect the ambiguity on the effects of knockdown of evectin-2 and ATP8A1, we included the following texts in the discussion (page 9, line 2↑).

“Knockdown of evectin-2 (an RE-resident protein) or ATP8A1 (an RE PS-flippase) inhibited the nuclear localization and signaling of YAP. Mechanistically, evectin-2 knockdown reduced the ubiquitination and increased the level of Lats1 that phosphorylates and inactivates YAP. ATP8A1 knockdown did not affect the amount of Lats1 or its ubiquitination, but increased the phosphorylated (activated) form of Lats1. We previously showed that, upon knockdown of ATP8A1, PS was still present in the cytosolic leaflet of RE membranes and that evectin-2 localized normally at REs⁸. Thus, ATP8A1 knockdown may not affect evectin-2 and therefore would not influence the ubiquitination or amounts of Lats1. At present, it is not clear how the concentrated PS in the cytosolic leaflet of RE membranes, which is generated by ATP8A1, affects the phosphorylation of Lats1.”